biophysics/microsystems/fluid mechanics

mocrofluidics, tree-like networks, dynamic response, Murray bifurcations

**Authors for correspondence:**
Rui D. M. Travasso
e-mail: ruit@uc.pt
Eugenia Corvera Poiré
e-mail: eugenia.corvera@gmail.com

# Resonances in the response of fluidic networks inherent to the cooperation between elasticity and bifurcations

Diana Yáñez[1], Rui D. M. Travasso[2] and Eugenia Corvera Poiré[1,2,3]

[1]Departamento de Física y Química Teórica, Facultad de Química, Universidad Nacional Autónoma de México, Ciudad de México 04510, Mexico
[2]CFisUC, Department of Physics, University of Coimbra, Rua Larga, Coimbra, 3004-516 Portugal
[3]Imaging Sciences and Biomedical Engineering Division, King's College, St Thomas' Hospital, London, UK

RDMT, 0000-0001-6078-0721; ECP, 0000-0002-4688-6922

A global response function (GRF) of an elastic network is introduced as a generalization of the response function (RF) of a rigid network, relating the average flow along the network with the pressure difference at its extremes. The GRF can be used to explore the frequency behaviour of a fluid confined in a tree-like symmetric elastic network in which vessels bifurcate into identical vessels. We study such dynamic response for elastic vessel networks containing viscous fluids. We find that the bifurcation structure, inherent to tree-like networks, qualitatively changes the dynamic response of a single elastic vessel, and gives resonances at certain frequencies. This implies that the average flow throughout the network could be enhanced if the pulsatile forcing at the network's inlet were imposed at the resonant frequencies. The resonant behaviour comes from the cooperation between the bifurcation structure and the elasticity of the network, since the GRF has no resonances either for a single elastic vessel or for a rigid network. We have found that resonances shift to high frequencies as the system becomes more rigid. We have studied two different symmetric tree-like network morphologies and found that, while many features are independent of network morphology, particular details of the response are morphology dependent. Our results could have applications to some biophysical networks, for which the morphology could be approximated to a tree-like symmetric structure and a constant pressure at the outlet. The GRF for these networks is a characteristic of the system fluid-network, being independent of the dynamic flow (or pressure) at the

network's inlet. It might therefore represent a good quantity to differentiate healthy vasculatures from those with a medical condition. Our results could also be experimentally relevant in the design of networks engraved in microdevices, since the limit of the rigid case is almost impossible to attain with the materials used in microfluidics and the condition of constant pressure at the outlet is often given by the atmospheric pressure.

# 1. Introduction

The presence of branched fluidic networks in nature is ubiquitous [1]. In plants, conducts called xylem transport the required water and nutrients from the roots to stems and leaves. These vessels present a space-filling branched structure capable of reaching all cells in the organism [2]. In mammals, blood flow carries oxygen from lungs and nutrients from the digestive system to all cells in the organism through vessel networks. Not surprisingly, several pathological scenarios associated with high mortality in humans are related to alterations in vascular networks that lead to changes in blood flow. When infarctions occur for example, vessel blockage stops blood irrigation to a tissue, leading to cell death. In the case of myocardial infarction, or heart attack, the lack of irrigation might lead to the impossibility of the heart to continue functioning [3]. In the case of cerebral infarction, if the brain cells lack oxygen for more than a few minutes they will die [4]. In aneurysm formation, deformation of vessel walls alter the flow profile and the stresses exerted by blood on the vessels, leading to vessel rupture [5,6]. Hypertension is mostly associated with vessel network rigidity [7–9]. Due to the importance of blood flow to human health, understanding how the characteristics of vessel networks lead to changes of the overall network flow is of the utmost importance. In particular, blood flow in the cardiovascular system has been modelled in great detail and from different perspectives [10–21]. Blood flow models can have different levels of detail depending on the phenomena they aim at probing. Complex three-dimensional Navier–Stokes calculations coupled with a detailed elastic treatment of vessel dynamics are able to model the flow in a blood cycle at every point within a vessel; however, their level of complexity make them prohibitive when studying blood flow in a large vascular network. In these situations zero- or one-dimensional models are convenient [13,17–20].

Several microfluidic devices are also constructed as branched networks of vessels, often with the aim of delivering fluid to a large number of sub-devices [22,23]. A characterization of how material properties, like elasticity, influence the global response function (GRF) in these branched microfluidic networks, permit better tailoring of the network morphologies and materials according to the need [24].

The response function (RF) of tree-like symmetric rigid networks, in which each vessel bifurcates into two equal vessels (not necessarily identical to the parent vessel), relates the flow that goes through the network with the pressure difference between the network extremes [25]. It is a concept extrapolated from the dynamic permeability of fluids confined in circular or rectangular channels [26–33], and it is essentially an effective permeability times an effective area [25]. Knowledge of the dynamic permeability in rigid channels, or of the RF in rigid networks, is important, since it allows one to know *a priori* the frequencies of the pressure signal that enhance the flow [30,31,33–36].

Several previous works of our group have shown that the RF depends strongly on the morphological properties of the network, on the rheological properties of the fluid and on the frequencies involved in the pressure pulse [34–36]. These works have been carried out on rigid vessel networks.

Recently, it has been found that the RF of a Newtonian fluid flowing in a single elastic vessel, that is able to deform along the flow direction, might have striking effects as a function of frequency in elastomeric materials at microscales [37]. A GRF for a tree-like symmetric elastic network, introduced as a generalization of the RF of a rigid network, relating the average flow along the network with the pressure difference at its extremes, has been introduced in the literature [38,39], but its behaviour as a function of frequency is yet to be studied.

In this paper, we study the GFR of tree-like symmetric elastic vessel networks, and explore the effect that the degree of elasticity and network morphology have on this frequency-dependent RF. Our study is relevant in microfluidic devices, where for a given pressure drop, flow rate in a deforming channel is found to be several times higher than the one expected in a non-deforming channel [40]. It could also be relevant for physiological vessel networks, since they are formed by elastic structures.

In §§2 and 3, for thoroughness of the presentation, we briefly describe a model for flow in elastic networks that has been published and validated for the arterial network [41]. In §2, we present the basic considerations to study flow in a single elastic vessel. In §3, we state the necessary

considerations to apply the model to elastic vessel networks. In §4, we introduce the global RF for tree-like symmetric elastic networks. In §5, we describe the two tree-like network morphologies that are used in this work. In §6, we find that the GRF is independent of the dynamics of the inflow, for networks that have constant pressure at the outlets, making the GRF a good quantity to study the network's dynamics. In §7, we show that the bifurcation structure of tree-like elastic networks causes the GRF to have resonances, which do not exist for rigid networks, nor for single elastic vessels. This implies that the flow magnitude across the network could be enhanced at certain frequencies due to the cooperation between the bifurcation structure and the elasticity of the network, via pulsatile forcing. We do a systematic study varying the networks' elasticity, and find features that are common to different network morphologies, and features that are morphology dependent. In §8, we present an analytical study of a single elastic bifurcation that demonstrates the emergence of the resonant behaviour. We present our conclusion in §9.

## 2. Flow through a single elastic vessel

To study the dynamics of each elastic vessel forming the network, we use the generalized Darcy's elastic model (GDEM) [41], which assumes the following:

First, a generalized Darcy's model for rigid cylindrical vessels, which gives a linear relation between flow, $\hat{q}$, and pressure gradient, $\partial \hat{p}/\partial x$, in frequency domain, is valid locally, that is, at any point $x$ along the flow direction,

$$\hat{q} = -\frac{AK(\omega)}{\eta}\frac{\partial \hat{p}}{\partial x}, \tag{2.1}$$

where $K(\omega) = -(\eta/i\omega\rho)[1 - 2\,J_1(\beta r)/\beta r J_0(\beta r)]$ stands for the dynamic permeability of a Newtonian fluid, which is a measurement of the resistance to flow; $J_0$ and $J_1$ are Bessel functions of the first kind of orders 0 and 1, respectively; $\beta^2 = i\rho\omega/\eta$, $r$ is the average radius of the elastic vessel; $\eta$ is the fluid viscosity; and $i = \sqrt{-1}$. Hats over quantities denote their Fourier transforms. Flow has been approximated as a velocity averaged over the average cross-sectional area, $A$, times this area.

Second, a linear relationship between pressure and flow gradient, coming from mass conservation and a Hooke-like linear model for vessel elasticity, is valid locally, namely

$$\hat{p} = \frac{1}{i\omega C}\frac{\partial \hat{q}}{\partial x}, \tag{2.2}$$

where $C = 3\pi r^3/2Eh$ is called the vessel compliance, $h$ is the vessel wall thickness, $E$ is the Young's modulus of the vessel given by

$$E = \frac{3\rho c^2 r}{2h}, \tag{2.3}$$

where $c$ is the pulse wave velocity. The limit of a rigid vessel is obtained when the vessel compliance vanishes, that is, when $C \to 0$.

Combining equations (2.1) and (2.2), one obtains a harmonic oscillator equation for the pressure in frequency domain, namely,

$$\frac{\partial^2 \hat{p}}{\partial x^2} = -\kappa^2 \hat{p}, \tag{2.4}$$

where $\kappa^2 = i\omega C\eta/AK(\omega)$.

Solving equation (2.4) for a single vessel of length $l$, with known $\hat{p}_{in}$ and $\hat{p}_{out}$ for the inlet and outlet pressures on the vessel extremes, respectively, yields an analytical expression for the pressure along the vessel in frequency domain given by

$$\hat{p}(x) = \hat{p}_{in}\cos(\kappa x) + \frac{\hat{p}_{out} - \hat{p}_{in}\cos(\kappa l)}{\sin(\kappa l)}\sin(\kappa x), \tag{2.5}$$

which, when substituted in equation (2.1), provides the flow along the vessel in frequency domain

$$\hat{q}(x) = M\left(\hat{p}_{in}\sin(\kappa x) - \frac{\hat{p}_{out} - \hat{p}_{in}\cos(\kappa l)}{\sin(\kappa l)}\cos(\kappa x)\right), \tag{2.6}$$

where $M^2 = i\omega C\,AK(\omega)/\eta$, and $x \in [0, l]$.

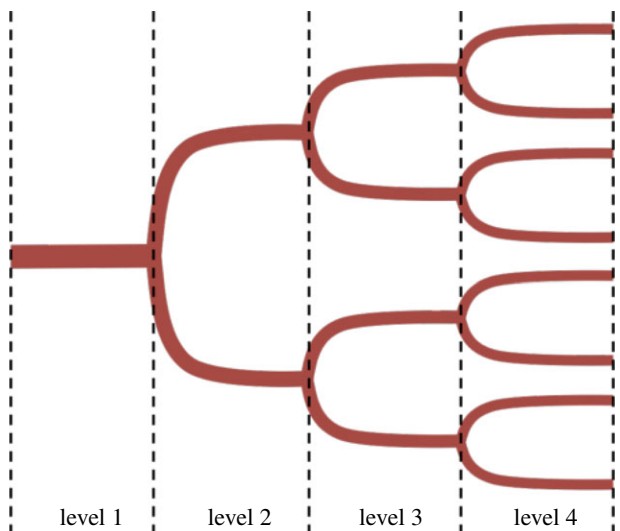

**Figure 1.** Illustration of a 4-level bifurcating tree-like network.

Solving equation (2.4) for a single vessel with known $\hat{q}_{\text{in}}$ and $\hat{p}_{\text{out}}$ for the inlet flow and outlet pressure on the vessel extremes, respectively, yields an analytic expression for the pressure along the vessel in frequency domain given by

$$\hat{p}(x) = \frac{\hat{q}_{\text{in}} \sin{(\kappa l)} + M\hat{p}_{\text{out}}}{M\cos{(\kappa l)}} \cos{(\kappa x)} - \frac{\hat{q}_{\text{in}}}{M} \sin{(\kappa x)}, \tag{2.7}$$

which, when substituted in equation (2.1), provides the flow along the vessel in frequency domain

$$\hat{q}(x) = \frac{\hat{q}_{\text{in}} \sin{(\kappa l)} + M\hat{p}_{\text{out}}}{\cos{(\kappa l)}} \sin{(\kappa x)} + \hat{q}_{\text{in}} \cos{(\kappa x)}. \tag{2.8}$$

# 3. Flow through an elastic vessel network

GDEM [41] considers elastic networks of tubes in which a viscous fluid flows subject to a time-dependent inlet flow (or pressure). In this paper, we focus on tree-like symmetric networks, in which cylindrical vessels bifurcate into two identical vessels, as illustrated in figure 1 for a 4-level network. The model deliberately excludes trifurcations or higher order branching, since bifurcations are the most probable and common forms of branching structures in nature. Articles in the literature that classify normal vascular networks consider only bifurcations [42–44]. More complex branching morphologies, such as trifurcations, are present in non-hierarchical pathological vasculatures, such as the ones irrigating tumours [45].

Tree-like networks will have two types of vessels: the ones in which boundary conditions are imposed, namely vessels that take inlet flows or pressures as boundary condition, or vessels in which outlet pressures are imposed; and vessels that are internal to the network. For these ones, the pressures at the vessels' extremes have to be determined as part of the solution.

Flow conservation is imposed at the nodes (points of bifurcation) as well as equal pressures on the extremes of the vessels connected to a node. For instance, for a vessel at level $j$, with length $l_j$, that bifurcates into vessels at level $j + 1$, with lengths $l_{j+1}$, flow conservation at the node joining these three vessels, requires the output flow at level $j$, $\hat{q}_j(x = l_j)$ to be equal to twice the input flow at level $j + 1$, $\hat{q}_{j+1}(x = 0)$, that is, $\hat{q}_j(x = l_j) = 2\hat{q}_{j+1}(x = 0)$. This allows one to write an equation, by means of equation (2.6) (or equation (2.8)), that involves: pressure (or flow) at the entrance of vessel at level $j$, pressure at the node, and pressures at the outlet ($x = l_{j+1}$) of vessels at level $j + 1$. For each node in the network, there will be an equation coming from flow conservation at that node. The result is a system of equations for the pressure at the nodes, that can be solved, analytically or numerically, using the methodology described in the literature [41].

Once the pressure at the nodes is known, pressure and flow as a function of the position, $x$, along any vessel of the network, can be obtained by making use of equations (2.5)–(2.8).

# 4. Global response for a tree-like symmetric elastic network

For a rigid network, the fact that flow is constant along the network allows one to describe the network behaviour in terms of a dynamic RF, $\chi(\omega)$, which is a measure of how much flow, $q_1(t)$, goes through the network, given a pressure difference, $\Delta p(t) = p_{out} - p_{in}$, between the network inlet and the network outlet [25]. In frequency domain, this can be written as

$$\hat{q}_1 = -\frac{\chi(\omega)\Delta\hat{p}(\omega)}{\eta L}, \tag{4.1}$$

where $L = \sum_i l_i$, is the sum of the vessels' length, $l_i$, at each level $i$. $\chi$ has been computed for a tree-like symmetric rigid network by means of an electrical analogy, adding resistances $l_i/A_iK_i$ in series or parallel, according to the network morphology [25], giving the following result:

$$\frac{1}{\chi} = \frac{1}{L}\sum_{i=1}^{N}\frac{l_i}{2^{i-1}A_iK_i}. \tag{4.2}$$

In equation (4.1), $\hat{q}_1$ is the flow at the inlet of the network, but given that flow is constant along a rigid network, it is also the total flow at any network level, $i$, containing $2^{i-1}$ vessels, that is, $\hat{q}_1 = 2^{i-1}\hat{q}_i$, where $\hat{q}_i$ is the flow in each of the vessels at level $i$.

For an elastic network, on the other hand, flow is not uniform along the network. Hence, we define a global response function (GRF), that would give information of the flow, integrated throughout the network, that is, the flow averaged along the flow direction, $\langle Q\rangle_x (t)$. A GRF for a tree-like symmetric elastic network is defined, in frequency domain, as [38,39]

$$\chi_{global} \equiv -\frac{\eta L\langle\hat{Q}\rangle_x}{\Delta\hat{p}}, \tag{4.3}$$

where

$$\langle\hat{Q}\rangle_x = \frac{1}{L}\sum_{i=1}^{N}\int_0^{l_i} 2^{i-1}\hat{q}_i(x)\,\mathrm{d}x = \frac{1}{L}\sum_{i=1}^{N} 2^{i-1}l_i\langle\hat{q}_i\rangle_x. \tag{4.4}$$

Here $\hat{q}_i(x)$ is the flow at position $x$ in one of the vessels at level $i$; $2^{i-1}\hat{q}_i(x)$ is the total flow at position $x$ in level $i$; and $\langle\hat{q}_i\rangle_x = (1/l_i)\int_0^{l_i}\hat{q}_i(x)\,\mathrm{d}x = -(A_iK_i/\eta l_i)\Delta\hat{p}_i$, is the average flow along a single vessel at level $i$. The second equality in this expression is obtained using equation (2.1). $\Delta\hat{p} = \sum_{i=1}^{N}\Delta\hat{p}_i$. Note that for a rigid network, $\langle\hat{Q}\rangle_x$ reduces to $\hat{q}_1$, which is the flow at point $x$, of the network's cross-sectional area. In the limit of a rigid network, equation (4.3) reduces to equation (4.1), with $\chi_{global} = \chi$, given by equation (4.2).

# 5. Network morphologies

In order to consider the impact that network morphology has on the dynamics, we study two different tree-like symmetric networks.

One of them, already used in previous works [25,34], approximates the morphological properties of the vasculature of a dog. The vessel morphology of the dog's circulatory system is described in detail in the literature [46], and we have approximated the number of vessels, lengths and radii to the closest numbers that satisfy branching into identical vessels (table 1). For convenience in the discussion, we will refer to this network as the 'dog's network'.

We also analyse networks whose vessels follow Murray's Law [49], since physiological studies have validated the agreement of Murray's Law in vascular systems of mammals [50] and even in networks that transport water in plants [51]. According to Murray's Law, when a vessel of radius $r_p$ bifurcates into two identical vessels, the next level vessels are narrower, with radii $r_d$, that obey $r_p^3 = 2r_d^3$. Therefore, the radii of vessels at level $i$, are related to the inlet vessel radius, $r_1$, by

$$r_i = \left(\frac{1}{2}\right)^{(i-1)/3} r_1. \tag{5.1}$$

For a Murray's network starting with the dimensions of the dog's aorta and containing the same number of levels as the 'dog's network', the radii of all levels is prescribed by equation (5.1). We have adjusted a

**Table 1.** Number and characteristics of vessels for the different levels of the dog's network. Taken from [25], and based on the anatomical measurements collected in [46]. Typical dimensions of vessel 1 are those of the aorta, typical dimensions of vessels 2–5 are those of large arteries; of vessels 6–9 are those of main arterial branches, of vessels 10–11 are those of terminal branches, of vessels 12–25 are those of arterioles and of vessels 26–29 are those of capillaries. The vessel wall width, $h$, was taken to be equal to $h = 0.1r$, where $r$ is the radius. Values used for the fluid viscosity and density were $\eta = 5.0 \times 10^{-3}$ kg m$^{-1}$ s$^{-1}$ and $\rho = 1050$ kg m$^{-3}$, respectively [47]. The Young's modulus $E$ is given by equation (2.3). For an arterial tree, the pulse wave velocity, $c$, is given by the empirical relationship $c = 13.3/(2\,r)^{0.3}$ (in m s$^{-1}$), with $r$ measured in millimetres [48]. This gives the values of $E_0$ in the table.

| levels | no. of vessels | radius (μm) | length (cm) | $E_0$ (MPa) |
|---|---|---|---|---|
| 1 | 1 | 5000 | 40.0 | 0.70 |
| 2–5 | 30 | 1500 | 20.0 | 1.4 |
| 6–9 | 480 | 500 | 10.0 | 2.8 |
| 10–11 | 1536 | 300 | 1.0 | 3.8 |
| 12–25 | 33 552 354 | 10 | 0.2 | 30 |
| 26–29 | 503 316 480 | 4 | 0.1 | 50 |

**Table 2.** Number and characteristics of vessels for the different levels of Murray's network. In this network vessel, radii and lengths are obtained as a function of level $n$. The vessel Young's modulus is a function of its radius $r$ (measured in micrometres). The vessel wall width, $h$, was taken to be equal to $h = 0.1r$ in all cases. Values for fluid viscosity and density are as in table 1. The Young's modulus $E$ is given by equation (2.3). For an arterial tree, the pulse wave velocity, $c$, is given by the empirical relationship $c = 13.3/(2r)^{0.3}$ (in m s$^{-1}$), with $r$ measured in millimetres [48]. This gives the expression for $E_0$ shown in the table, with $r$ measured in millimetres.

| levels | radius (μm) | length (cm) | $E_0$ (MPa) |
|---|---|---|---|
| 1 | 5000 | 40.0 | 0.70 |
| 2–28 | $5000/2^{(n-1)/3}$ | $40\,n^{-1.78}$ | $1.84\,r^{-0.6}$ |
| 29 | 7.8 | 0.1 | 34 |

power law for the lengths [35]. Parameters used in the calculations are reported in table 2. The caption of table 1 includes the fluid parameters used in this work.

# 6. Independence on boundary conditions

For a network response like the one in equation (4.3) to be useful, it must prove to be independent on boundary conditions. We limit the study to the case of networks whose pressure is constant at the outlet, and consider three different inflows as incident boundary condition. Namely, an aortic physiological flow measured *in vivo* [52], a simple time-dependent one-mode cosine function, and a simple pulse consisting of a delta function at time zero. We have chosen a constant pressure at the network outlet of 4.2 kPa for the three calculations, a value that is consistent with physiological situations. Following the methodology sketched out in §3, and explained in detailed in reference [41], we have computed flows and pressures all along the network. We have used them to compute the RF defined in equation (4.3). We have verified that the RFs of the network, using these three different inflows, and constant pressure at the network outlet, are identical. Since, by construction, the inflow consisting of a simple pulse in time domain, is constant in frequency domain, it has less noise than the other signals, and it is the one reported in all figures. We have also verified that halving or doubling the inflows or the outlet pressures do not alter the RF.

# 7. Results

Figure 2 shows the absolute value of the RF for two different networks. The first thing to notice is that both responses are non-monotonic functions of frequency and therefore present resonances (maxima at

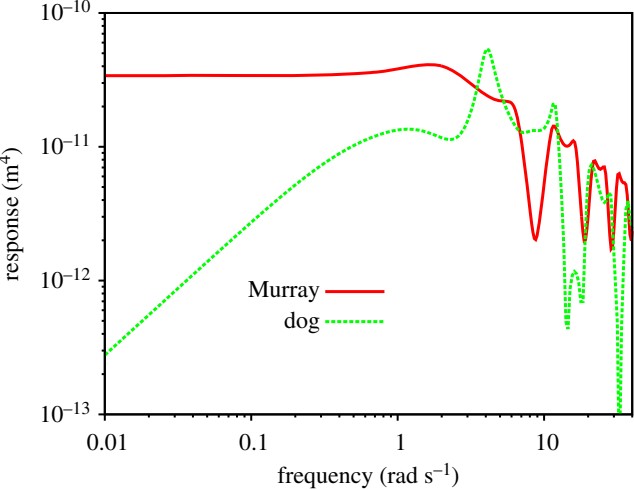

**Figure 2.** Magnitude of the RF as a function of frequency for the dog's and Murray's elastic networks for 'physiological' Young's moduli, $E_0$. Parameters for the calculation are given in tables 1 and 2.

certain frequencies). This implies that the flow magnitude would be enhanced at certain frequencies of the pulsatile forcing. This enhancement comes from the cooperation between the bifurcation structure and the elasticity of the network via pulsatile forcing. Elasticity by itself does not cause resonances in the flow of Newtonian fluids subject to pulsatile forcing. That is, for a single vessel, the RF, $\chi_{global}$, given by equation (4.3), decreases monotonically with frequency (this will be explicitly shown in §8). The structure of bifurcations is necessary to have resonances.

There is an interplay between network geometry and network elasticity to give the exact location of resonances. In figure 2, we can observe that the main resonance (the one with the highest value of the RF) for Murray's network occurs around $\omega = 2$ rad s$^{-1}$, while for the dog's network occurs around $\omega = 4$ rad s$^{-1}$. Values of the RF at resonance are $4 \times 10^{-11}$ m$^4$ and $5 \times 10^{-11}$ m$^4$, respectively. These values are surprisingly close to each other, given that the steady-state network responses (at low frequencies, out of the range of figure 2) are five orders of magnitude different. This is because the outer, wider vessels, which are of a similar calibre in both networks, determine the high-frequency behaviour of the response. The elastic properties of wider vessels allow for better accumulation and release of fluid during oscillatory flow, causing the non-monotonic high-frequency behaviour of the RF.

The presence of vessel elasticity changes qualitatively the dynamic response of rigid networks, for which there are no resonances. Figure 3 shows how the RF changes as the Young's moduli of the networks, $E$, are varied as multiples of $E_0$. For both networks, Murray's in figure 3a and the dog's one in figure 3b, resonances shift to high frequencies as the system becomes more rigid. For both networks, resonance frequencies are proportional to the square root of the network's Young's moduli, that is, $\omega_{res} \sim (E/E_0)^{1/2}$ (see §8). Resonance frequencies as a function of network's elasticity are shown in red in figure 4a for Murray's network. The continuous line, shown for reference, has a slope of 1/2. For the dog's network, frequencies for the first two maxima are shown in figure 4b. In between the points indicating the frequencies of the first and second maxima, there is a continuous line, shown for reference, with a slope of 1/2. As the network becomes more rigid, the first maximum takes over and becomes the main resonance of the system. The main resonance is plotted in red.

We have also found that the magnitude of the responses at resonance decreases for increasing network rigidity. This can be appreciated in figure 3a. Resonances disappear, for large network rigidities, when the values of the response function, at all frequencies, are below the value of the response at zero frequency (not shown). These are features of the RF that are independent of the network morphologies studied. Particular details of the RF are morphology-dependent. For Murray's network (figure 3a), there is a frequency region around the first resonance where the value of the response is larger than for a rigid network, while the high-frequency behaviour presents responses below the one of the rigid network and the low-frequency behaviour is quite similar to the one of the rigid network. By contrast, for the dog's network (figure 3b), vessel elasticity changes dramatically the low-frequency behaviour of the response, causing the network response to increase as a function of frequency. This is because inner, thinner vessels (with high resistance and small permeability), which are much smaller for the dog's network than for the Murray's network, determine the low-frequency

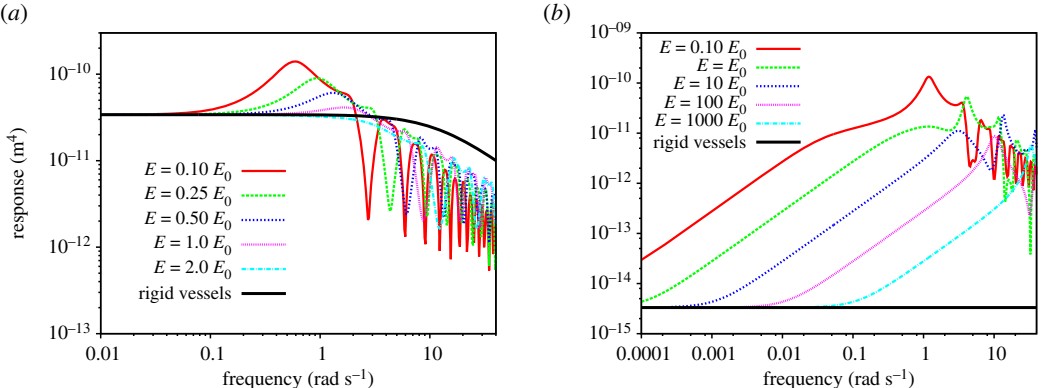

**Figure 3.** (*a*) Magnitude of the RF as a function of frequency for Murray's elastic networks with various degrees of elasticity. (*b*) Magnitude of the RF as a function of frequency for the dog's elastic network with various degrees of elasticity. In both figures, the magnitude of the response of the corresponding rigid network is shown in black.

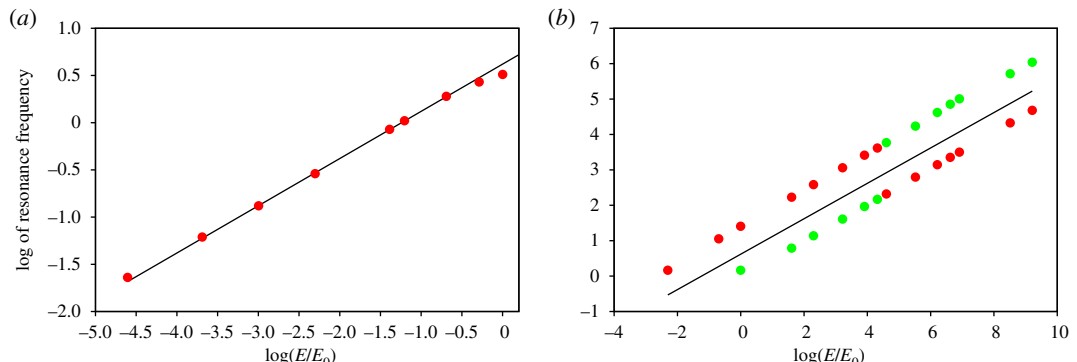

**Figure 4.** (*a*) Resonance frequency for a Murray's network as a function of network's elasticity. (*b*) Frequencies for the first two maxima of the dog's network as a function of network's elasticity. As the network becomes more rigid, the first maximum takes over and becomes the dominant resonance of the system. The dominant resonance is plotted in red. Continuous black lines in both figures have a slope of 1/2 and are shown for reference.

behaviour of the RF. To accommodate the low-frequency behaviour of the response (given by the small vessels, with large resistance and low permeability) and the high-frequency behaviour of the response (given by the outer wider vessels, with low resistance and high permeability), the network's response increases as a function of frequency. This behaviour is qualitatively different from the monotonic decrease of the RF for rigid networks. Also, for the dog's network, the value of the response increases by several orders of magnitude, for most of the frequency spectrum, relative to the one of a rigid network. This implies that the maximum amplitudes of the overall flow, regardless of the frequencies involved in the pressure drop across the network, would increase significantly with respect to the rigid case.

# 8. Origin of resonances and scaling behaviour

As stated in §7, we have found resonance frequencies for the GRF of a Newtonian fluid in an elastic network, using a model that for a Newtonian fluid in a single elastic vessel does not exhibit resonances. This implies that the resonant behaviour is due to the structure of bifurcations, inherent to tree-like networks. In order to better see this, we compute analytically the GRF for a network consisting of a single bifurcation, and show the emergence of the non-monotonic behaviour as a function of frequency.

The response for a Newtonian fluid in a single elastic vessel, with average cross-sectional area, $A$, can be obtained analytically from the equation for flow along the vessel (equation (2.6)) that, when integrated along the flow direction, gives

$$\langle \hat{q} \rangle_x = -\frac{AK(\omega)}{\eta l}(\hat{p}_{\text{out}} - \hat{p}_{\text{in}}),$$ (8.1)

which gives an RF, $\chi_{\text{global}}$, defined in equation (4.3), that is equal to the area, $A$, times the dynamic permeability, $K(\omega)$, i.e. $\chi_{\text{global}} = A\, K(\omega)$, with $K(\omega)$ given by the analytical expression after equation (2.1) and, for a Newtonian fluid, has a monotonic decay as a function of frequency [26,31].

We now consider a network with a single bifurcation and compute its RF. We consider a vessel at level 1, that bifurcates into two identical vessels at level 2. Flow conservation at the bifurcation implies that outflow in vessel at level 1, is equal to the sum of inflows of vessels after the bifurcation, namely $\hat{q}_1(x = l_1) = 2\hat{q}_2(x = 0)$. For vessel at level 1, the pressure at the vessel's entrance, is the pressure at the entrance to the system, $\hat{p}_{\text{in}}$, and the pressure at the exit is an unknown pressure at the node, $\hat{p}_N$. For vessels at level 2, the pressure at the entrance is the pressure at the node, $\hat{p}_N$, and the pressure at the vessels' exit is the output pressure of the system, $\hat{p}_{\text{out}}$. Using equation (2.6), in the equation for flow conservation at the node, we can write an equation for the pressure at the node, $\hat{p}_N$. This one is given by

$$\hat{p}_N = \frac{2A_2 K_2 \kappa_2 \sin(\kappa_1 l_1)\hat{p}_{\text{out}} + A_1 K_1 \kappa_1 \sin(\kappa_2 l_2)\hat{p}_{\text{in}}}{A_1 K_1 \kappa_1 \sin(\kappa_2 l_2)\cos(\kappa_1 l_1) + 2A_2 K_2 \kappa_2 \sin(\kappa_1 l_1)\cos(\kappa_2 l_2)}. \tag{8.2}$$

On the other hand (using equation (4.4)), the average flow along the network with a single bifurcation is

$$\langle \hat{Q} \rangle_x = \frac{1}{l_1 + l_2}(l_1 \langle \hat{q}_1 \rangle_x + 2l_2 \langle \hat{q}_2 \rangle_x), \tag{8.3}$$

which, using equation (8.1), can be written as

$$\langle \hat{Q} \rangle_x = -\frac{(2A_2 K_2 \hat{p}_{\text{out}} - A_1 K_1 \hat{p}_{\text{in}})}{\eta(l_1 + l_2)} - \frac{(A_1 K_1 - 2A_2 K_2)}{\eta(l_1 + l_2)}\hat{p}_N, \tag{8.4}$$

with $\hat{p}_N$ given by equation (8.2). Accordingly, the RF, defined in equation (4.3), is given by

$$\chi_{\text{global}} = \frac{(2A_2 K_2 \hat{p}_{\text{out}} - A_1 K_1 \hat{p}_{\text{in}}) + (A_1 K_1 - 2A_2 K_2)\hat{p}_N}{\hat{p}_{\text{out}} - \hat{p}_{\text{in}}}, \tag{8.5}$$

From this expression, it becomes clear why there is a non-monotonic behaviour as a function of frequency coming from the bifurcation. That is, the pressure at the node, $\hat{p}_N$, needed to compute $\chi_{\text{global}}$, and given by equation (8.2), contains non-monotonic sinusoidal terms in $\kappa_1 l_1$ and $\kappa_2 l_2$, that are functions of frequency and of the mechanical properties of the vessels (see expression for $\kappa$ after equation (2.4)). These terms were not averaged out when the integration along the flow was performed, as happened for a single vessel.

Figure 5 illustrates the RF for a network, that consists of a single bifurcation, with the characteristics of the first three vessels for both the dog's and Murray's networks. We have chosen a zero pressure at the outlet. The results clearly show the non-monotonic behaviour coming from the bifurcation. For this particular example,

$$\chi_{\text{global}} = A_1 K_1 - (A_1 K_1 - 2A_2 K_2)\frac{\hat{p}_N^*}{\hat{p}_{\text{in}}}, \tag{8.6}$$

where

$$\frac{\hat{p}_N^*}{\hat{p}_{\text{in}}} = \frac{A_1 K_1 \kappa_1 \sin(\kappa_2 l_2)}{A_1 K_1 \kappa_1 \sin(\kappa_2 l_2)\cos(\kappa_1 l_1) + 2A_2 K_2 \kappa_2 \sin(\kappa_1 l_1)\cos(\kappa_2 l_2)}. \tag{8.7}$$

For this example, the RF, $\chi_{\text{global}}$, is explicitly independent from the pressure at the inlet. It is straightforward to prove that, for constant $p_{\text{out}}(t)$, $\chi_{\text{global}}$ is independent of the pressure drop.

In order to understand the scaling behaviour observed in figure 4, we notice that the GRF has terms in $K_i \kappa_i$, which represent slowly varying modes of frequency, and terms in $\cos(\kappa_i l_i)$ and $\sin(\kappa_i l_i)$, which are rapid modes, that will determine the GRF extremes, and therefore the resonances. From the expressions for $K$, $C$ and $\kappa$ (after equations (2.1), (2.2) and (2.4)), and since $\kappa l$ is a non-dimensional quantity, we can obtain a characteristic frequency of each vessel in the system, given by $\omega_i = (1/l_i)(E_i h_i/\rho r_i)^{1/2}$. As in many elastic systems, for instance, a forced harmonic oscillator, resonances appear when the forcing is made at the smallest characteristic frequency of the system. In this case, the frequency characteristic of the largest vessel in the network. This also explains why resonances shift to high frequencies as the system becomes more rigid (with higher values of Young's moduli).

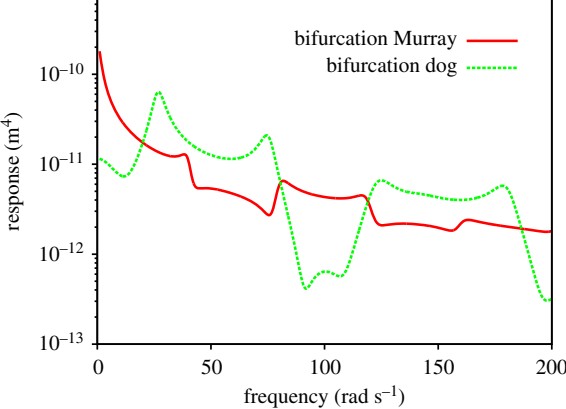

**Figure 5.** Magnitude of the RF for a single bifurcation.

# 9. Conclusion

A GRF of a tree-like symmetric elastic network is introduced as a generalization of the RF of a rigid network [38,39]. The GRF relates the network's flow, averaged along the flow direction, with the pressure difference at the network's extremes. It can be used to explore the frequency behaviour of a fluid confined in an elastic network. The GRF indicates which frequencies, involved in the dynamic pressure drop, maximize the magnitude of flow averaged along the flow direction. We have found resonance frequencies of the GRF for Newtonian fluids in elastic networks using a model that for a single elastic vessel, and for rigid networks, does not give resonances, and proved that this resonant behaviour is due to the cooperation between elasticity and bifurcations.

Some of the features of the GRF are common to networks of different morphologies, for instance, for all networks, resonance frequencies shift to high frequencies as the system becomes more rigid. For all of them, responses at resonance decrease for increasing network rigidity. Particular details of the RF are morphology-dependent. For example, in the dog's network studied here, vessel elasticity changes dramatically the low-frequency behaviour of the GRF, causing this one to increase as a function of frequency. This behaviour could be experimentally important for certain networks engraved in microdevices, since the limit of the rigid case is almost impossible to attain with the materials used in microfluidics.

For networks in which pressure is constant at the outlets, the GRF is characteristic of the system fluid-network, and independent of the dynamics of the inflow and of the value of pressure at the network's outlet. It might therefore represent a good quantity to differentiate healthy vasculatures from those with a medical condition. Abnormalities in large vessels could possibly be observed in the high-frequency behaviour of the GRF, while abnormalities in small vessels would in principle be observable in the low-frequency behaviour of the GRF. Whether or not this quantity might be clinically relevant to discriminate vasculatures with a medical condition, from those of a control group, is yet to be explored.

Our methodology could also be applicable to the domain of microfluidics, where branched symmetric structures are often engraved in microchips whose materials range from elastomeric to rigid. For a possible experimental verification of our results, it would be worth recalling that, for given values of the elastic and fluid parameters of a microfluidic device, one can always attain the linear flow regime by decreasing the amplitude of the dynamic pressure drop.

Data accessibility. The results presented in the figures are the direct calculations of the respective equations for the RFs.
Authors' contributions. D.Y., R.D.M.T. and E.C. performed the analytical and numerical calculations. R.D.M.T. and E.C.P. wrote the article. E.C.P. coordinated the work. All authors gave final approval for publication.
Competing interests. We declare we have no competing interests.
Funding. D.Y. and E.C.P. thank funding from CONACyT (Mexico), through project no. 219584, and the Faculty of Chemistry UNAM, through *subprograma* 127. R.D.M.T. thanks the support of FEDER funds through the Operational Program Competitiveness Factors – COMPETE and to national funds by FCT – Foundation for Science and Technologyunder the strategic project no. UID/FIS/04564/2016 and under POCI-01-0145-FEDER-031743 – PTDC/BIA-CEL/31743/2017. E.C.P. thanks funding from CONACyT (Mexico) through agreement no. 2018-000007-01EXTV-00183; and from DGAPA, UNAM through a PASPA programme, during a sabbatical leave.

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
