## [Reviewer comments · Royal Society Open Science]

Review History

RSOS-190661.R0 (Original submission)

Review form: Reviewer 1

Is the manuscript scientifically sound in its present form?

Yes

Are the interpretations and conclusions justified by the results?

Yes

Is the language acceptable?

Yes

Is it clear how to access all supporting data?

No

Do you have any ethical concerns with this paper?

No

Have you any concerns about statistical analyses in this paper?

No

Recommendation?

Accept with minor revision (please list in comments)

Comments to the Author(s)

See attached (Appendix A).

Review form: Reviewer 2

Is the manuscript scientifically sound in its present form?

Yes

Are the interpretations and conclusions justified by the results?

No

Is the language acceptable?

No

Is it clear how to access all supporting data?

Not Applicable

Do you have any ethical concerns with this paper?

No

Have you any concerns about statistical analyses in this paper?

No

Recommendation?

Reject

Comments to the Author(s)

Major points.

This submission on networks of modelled elastic vessels with viscous fluid flow has interesting aspects to it, but there are some flaws.

The subject has been studied previously by many authors whose work is missed out here throughout the paper. The novelty appears debatable. The fact that branching alters the system response significantly is well known. Possible enhancement of flow is established. Papers in, for example, the Journal of Engineering Mathematics from 2003 to the present would enlarge the background considerably - authors and/or relevant references include Secomb, Sherwin, Balta, Wu, Pries, van de Vosse, Pedley, Quarteroni, Olufsen, among others.

Having just two different morphologies (figures 2, 3, 5 etc and in the corresponding text) seems inadequate to determine a general trend or hard conclusions.

Much material is simply repeated in the abstract, at the end of the Introduction and in the Conclusions. Also, generally, the presentation is far too lengthy given what has been done in previous research.

A line of slope $\frac{1}{2}$ provides a promising comparison in figure 4 and in the accompanying text. Yet the obvious question of why $\frac{1}{2}$ is the appropriate slope is not addressed in the paper. Similarly the resonance shift to high frequencies remains not only unexplained but hardly considered.

The presentation seems disorganised and inconsistent in parts. There are conjectures but no serious links with experiments. Fluid parameters are mentioned in the section heading on page 9 but appear to be absent in that section.

Minor points.

There are numerous errors of grammar scattered through the paper.

Review form: Reviewer 3

Is the manuscript scientifically sound in its present form?

Yes

Are the interpretations and conclusions justified by the results?

Yes

Is the language acceptable?

Yes

Is it clear how to access all supporting data?

Not Applicable

Do you have any ethical concerns with this paper?

No

Have you any concerns about statistical analyses in this paper?

No

Recommendation?

Accept with minor revision (please list in comments)

Comments to the Author(s)

In my opinion, the paper is correct, well explained, and with interesting results. The physical discussion of the results are clear and well stated.

This work analyzes the dynamic response of an elastic tube network saturated with a viscous fluid. The main results indicate that this dynamic response of the network differs from the dynamic response of a single tube. They found that the previous decreasing monotonic behavior of the resonant frequency is no longer always valid. In terms of the impact, I can see possible applications in terms of biophysical applications and microdevices with fluids inside.

The authors analyze, from the analytical point of view, through mathematical modeling of momentum transport equations, the problem of the dynamic response of a single tube or tube networks. The model is well constructed, and the assumptions have been established.

In previous work of the authors, they found allometric relations in terms of the longer diameter of the initial (wider) tube of the networks. The authors may give a discussion of the possible similar results in this case, where the elastic property of the tube is crucial.

Decision letter (RSOS-190661.R0)

26-Jun-2019

Dear Dr Travasso,

The editors assigned to your paper ("Resonances in the response of fluidic elastic networks inherent to bifurcations") have now received comments from reviewers. We would like you to revise your paper in accordance with the referee and Associate Editor suggestions which can be found below (not including confidential reports to the Editor). Please note this decision does not guarantee eventual acceptance.

Please submit a copy of your revised paper before 19-Jul-2019. Please note that the revision deadline will expire at 00.00am on this date. If we do not hear from you within this time then it will be assumed that the paper has been withdrawn. In exceptional circumstances, extensions may be possible if agreed with the Editorial Office in advance. We do not allow multiple rounds of revision so we urge you to make every effort to fully address all of the comments at this stage. If deemed necessary by the Editors, your manuscript will be sent back to one or more of the original reviewers for assessment. If the original reviewers are not available, we may invite new reviewers.

- Data accessibility

It is a condition of publication that all supporting data are made available either as supplementary information or preferably in a suitable permanent repository. The data accessibility section should state where the article's supporting data can be accessed. This section

should also include details, where possible of where to access other relevant research materials such as statistical tools, protocols, software etc can be accessed. If the data have been deposited in an external repository this section should list the database, accession number and link to the DOI for all data from the article that have been made publicly available. Data sets that have been deposited in an external repository and have a DOI should also be appropriately cited in the manuscript and included in the reference list.

If you wish to submit your supporting data or code to Dryad (<http://datadryad.org/>), or modify your current submission to dryad, please use the following link:
<http://datadryad.org/submit?journalID=RSOS&manu=RSOS-190661>

- **Competing interests**

- **Authors' contributions**

- **Acknowledgements**

- **Funding statement**

Kind regards,

on behalf of Dr Peter Stewart (Associate Editor) and R. Kerry Rowe (Subject Editor)
openscience@royalsociety.org

Associate Editor's comments (Dr Peter Stewart):

Comments to the Author:

We have now received three referee reports on your paper. Of these, Reviewers 1 and 3 recommend only minor revisions. However, Reviewer 2 has more substantial reservations, primarily related to the novelty of the results presented in the paper.

In response to these comments, we recommend that the authors undertake a major revision of the manuscript to address the issues raised by each of the three referees.

Reviewers' Comments to Author:

Reviewer 1:

See attached file

Reviewer 2:

Major points.

This submission on networks of modelled elastic vessels with viscous fluid flow has interesting aspects to it, but there are some flaws.

The subject has been studied previously by many authors whose work is missed out here throughout the paper. The novelty appears debatable. The fact that branching alters the system response significantly is well known. Possible enhancement of flow is established. Papers in, for example, the Journal of Engineering Mathematics from 2003 to the present would enlarge the background considerably - authors and/or relevant references include Secomb, Sherwin, Balta, Wu, Pries, van de Vosse, Pedley, Quarteroni, Olufsen, among others.

Having just two different morphologies (figures 2, 3, 5 etc and in the corresponding text) seems inadequate to determine a general trend or hard conclusions.

Much material is simply repeated in the abstract, at the end of the Introduction and in the Conclusions. Also, generally, the presentation is far too lengthy given what has been done in previous research.

A line of slope $\frac{1}{2}$ provides a promising comparison in figure 4 and in the accompanying text. Yet the obvious question of why $\frac{1}{2}$ is the appropriate slope is not addressed in the paper. Similarly the resonance shift to high frequencies remains not only unexplained but hardly considered.

The presentation seems disorganised and inconsistent in parts. There are conjectures but no serious links with experiments. Fluid parameters are mentioned in the section heading on page 9 but appear to be absent in that section.

Minor points.

There are numerous errors of grammar scattered through the paper.

Reviewer 3:

In my opinion, the paper is correct, well explained, and with interesting results. The physical discussion of the results are clear and well stated.

This work analyzes the dynamic response of an elastic tube network saturated with a viscous fluid. The main results indicate that this dynamic response of the network differs from the dynamic response of a single tube. They found that the previous decreasing monotonic behavior of the resonant frequency is no longer always valid. In terms of the impact, I can see possible applications in terms of biophysical applications and microdevices with fluids inside.

The authors analyze, from the analytical point of view, through mathematical modeling of momentum transport equations, the problem of the dynamic response of a single tube or tube networks. The model is well constructed, and the assumptions have been established.

In previous work of the authors, they found allometric relations in terms of the longer diameter of the initial (wider) tube of the networks. The authors may give a discussion of the possible similar results in this case, where the elastic property of the tube is crucial.

Author's Response to Decision Letter for (RSOS-190661.R0)

See Appendices B & C.

Decision letter (RSOS-190661.R1)

21-Aug-2019

Dear Dr Travasso:

On behalf of the Editors, I am pleased to inform you that your Manuscript RSOS-190661.R1 entitled "Resonances in the response of fluidic networks inherent to the cooperation between elasticity and bifurcations" has been accepted for publication in Royal Society Open Science subject to minor revision in accordance with the Editor's suggestions at the end of this email.

The Editors have recommended publication, but also suggest some minor revisions to your manuscript. Therefore, I invite you to respond to the comments and revise your manuscript.

- Ethics statement

- Data accessibility

<http://datadryad.org/submit?journalID=RSOS&manu=RSOS-190661.R1>

- Competing interests

- Authors' contributions

- Acknowledgements

- Funding statement

Because the schedule for publication is very tight, it is a condition of publication that you submit the revised version of your manuscript before 30-Aug-2019. Please note that the revision deadline will expire at 00.00am on this date. If you do not think you will be able to meet this date please let me know immediately.

To revise your manuscript, log into <https://mc.manuscriptcentral.com//rsos> and enter your Author Centre, where you will find your manuscript title listed under "Manuscripts with Decisions". Under "Actions," click on "Create a Revision." You will be unable to make your revisions on the originally submitted version of the manuscript. Instead, revise your manuscript and upload a new version through your Author Centre.

on behalf of Dr Peter Stewart (Associate Editor) and R. Kerry Rowe (Subject Editor)
openscience@royalsociety.org

Associate Editor Comments to Author (Dr Peter Stewart):

Page 3, use referencing style as per journal []

Page 3, line 50 'may permit to better tailor the' -> 'permit better tailoring of the'

Page 3, line 55 use of identical is ambiguous - are the daughter vessels identical to the parent as well as to each other?

Page 4, line 15 'done' -> 'carried out on'

Page 4, line 28 - do not use references in a sentence. Several other instances through the manuscript.

Page 5. Avoid use of bullet points.

Page 6. Why are you using the pulse wave velocity? Is this not just a wavespeed?

Page 7 Line 24 should read 'Articles in the literature that classify normal vascular networks consider only bifurcations.'

Page 8, line 33, should the q's not have hats as in Eq (9)

Page 8, use \langle and \rangle instead of $<$ and $>$

Author's Response to Decision Letter for (RSOS-190661.R1)

See Appendix D.

Decision letter (RSOS-190661.R2)

27-Aug-2019

Dear Dr Corvera,

I am pleased to inform you that your manuscript entitled "Resonances in the response of fluidic networks inherent to the cooperation between elasticity and bifurcations" is now accepted for publication in Royal Society Open Science.

Kind regards,
Lianne Parkhouse
Editorial Coordinator
Royal Society Open Science
openscience@royalsociety.org

on behalf of Dr Peter Stewart (Associate Editor) and R. Kerry Rowe (Subject Editor)
openscience@royalsociety.org

Appendix A

‘Resonances in the response of fluidic elastic networks inherent to bifurcations’

by Yañez, Travasso and Corvera Poiré

Outline of paper

The paper instigates the effect of elasticity of vessel walls on the flow of a viscous fluid through a bifurcating network of vessels. The results were performed on two differing model structures, but both produced resonances in the flow rate responses for particular system frequencies.

I am not an expert in this research field, but in my opinion the work presented is sound, and well written, hence I recommend publication. Below are some minor points the authors may wish to consider.

Comments

- Page 5, eqn (1): Is \hat{q} the flow rate? This should be made clear if it is.
- Page 5, line -2: ‘Bessel functions of the first kind’
- Page 5, line -1: What are η and i ? I assume they are viscosity and $\sqrt{-1}$ respectively. This should be noted.
- Page 6, eqn (5): This is valid for $x \in [0, l]$ which needs to be made clear.
- Page 7, Fig 1: Can you mark on so level numbers to this figure as this would help the non-expert reader.
- Page 7, line -2: Missing full stop.
- Page 8, eqn (9): L has been introduced without being defined. Is it the sum of all the l_i ?
- Page 8, line after (10): Remove second ‘in’.
- Page 10, halfway through section: You should consider numbering your sections and then referring to them by the section number in the text. This looks a lot less clumsy.

Appendix B

We thank the three referees for their comments, which have helped to improve the manuscript. We are sending a corrected version of the article where changes are marked in blue.

Reviewer #1

"The paper instigates the effect of elasticity of vessel walls on the flow of a viscous fluid through a bifurcating network of vessels. The results were performed on two differing model structures, but both produced resonances in the flow rate responses for particular system frequencies. I am not an expert in this research field, but in my opinion the work presented is sound, and well written, hence I recommend publication. Below are some minor points the authors may wish to consider."

"Page 5, eqn (1): Is \hat{q} the flow rate? This should be made clear if it is."

It is the Fourier transform of the flow rate. It is stated in the document just before Eq(1).

"Page 5, line -2: 'Bessel functions of the first kind'"

Corrected.

"Page 5, line -1: What are η and i ? I assume they are viscosity and $\sqrt{-1}$ respectively. This should be noted."

Corrected.

"Page 6, eqn (5): This is valid for $x \in [0, \ell]$ which needs to be made clear."

Corrected.

"Page 7, Fig 1: Can you mark on so level numbers to this figure as this would help the non-expert reader."

Done.

"Page 7, line -2: Missing full stop."

Corrected.

"Page 8, eqn (9): L has been introduced without being defined. Is it the sum of all the l_i ?"

Indeed, it is now clear after Eq 9.

"Page 8, line after (10): Remove second 'in'."

Removed.

"Page 10, halfway through section: You should consider numbering your sections and then referring to them by the section number in the text. This looks a lot less clumsy."

Done.

Reviewer #2:

"The subject has been studied previously by many authors whose work is missed out here throughout the paper. The novelty appears debatable. The fact that branching alters the system response significantly is well known. Possible enhancement of flow is established. Papers in, for example, the Journal of Engineering Mathematics from 2003 to the present would enlarge the background considerably - authors and/or relevant references include Secomb, Sherwin, Balta, Wu, Pries, van de Vosse, Pedley, Quarteroni, Olufsen, among others."

We invite the referee to follow large amount of alterations and calculations made to answer his/her concerns and suggestions. In this document we will mention our reasoning but we will not detail the many alterations marked in blue in the manuscript.

We partially agree with the referee. Studies of fluids in networks are ubiquitous. However, we think that the referee overlooked the novelty of our manuscript. First, we are not only addressing blood vasculatures or models of the arterial system. We hope with this new version to clarify that we are studying fluid flow in tree-like symmetric networks. The conclusions of this work can be applied to particular vascular networks, but also to microfluidic devices. The novelty of our paper consists of having found that resonances in the response function, defined in a way that it gives a measurement of the flow magnitude, averaged along the flow direction, can appear as the result of the cooperation between the network elasticity and the presence of bifurcations. This is much more specific than the well-known fact that branching "alters" the system response (in a sense that it produces changes in fluid flow). Also, the referee states that: *"Possible enhancement of flow is established"*, it seems that again, the referee overlooked that our statement is very precise: It is observed an enhancement of the flow averaged along the flow direction for certain frequencies of the pulsatile pressure gradient. We have done an extensive review of the literature, and this concept is new. Possibly, the referee was thinking on the well-known fact that when elasticity is present, flow can be enhanced or decreased in different parts of a network and different times. We hope that in this corrected version we succeeded in passing our message more clearly.

The literature of mathematical modeling of the arterial network is indeed very vast. We thank the reviewer for suggesting to include these references in the manuscript. Among the articles cited, there are also extensive reviews, which include an even more complete set of references to the topic. We again emphasize that our paper is not dealing with modeling of the arterial system. It is something more general, simpler, and we think, more fundamental than that.

We have added the following paragraph to the introduction:

In particular blood flow in the cardiovascular system has been modelled in great detail and from different perspectives [13–24]. Blood flow models can have different levels of detail depending on the phenomena they aim at probing. Complex 3D Navier-Stokes calculations coupled with a detailed elastic treatment of vessel dynamics are able to model the flow in a blood cycle at every point within a vessel, however their level of complexity make them prohibitive when studying blood flow in a large vascular network. In these situations 0D or 1D models are convenient [16,20–23].

And we have made clearer that we are dealing with tree-like symmetrical networks in which vessels bifurcate into identical vessels throughout the manuscript.

We have also added the following references.

10 F. N. Van de Vosse, "Mathematical modelling of the cardiovascular system", Journal of Engineering Mathematics 47, 175-183 (2003).

- 11 F. T. Smith, R. Purvis, S. C. R. Dennis, M. A. Jones, N. C. Ovenden, and M. Tadjfar, "Fluid flow through various branching tubes", *Journal of Engineering Mathematics*, 47, 277-298 (2003).
- 12 A. R. Pries, and T. W. Secomb, "Blood flow in microvascular networks", in *Microcirculation*, pp. 3-36, Academic Press (2008).
- 13 M. S. Olufsen, and A. Nadim, "On deriving lumped models for blood flow and pressure in the systemic arteries", in *Computational Fluid and Solid Mechanics*, pp. 1786-1789, Elsevier Science Ltd (2003).
- 14 Y. Gandica, T. Schwarz, O. Oliveira, and R. D. Travasso, "Hypoxia in vascular networks: a complex system approach to unravel the diabetic paradox", *PloS one*, 9, e113165 (2014).
- 15 B. E. Carlson, J. C. Arciero, and T. W. Secomb, "Theoretical model of blood flow auto-regulation: roles of myogenic, shear-dependent, and metabolic responses", *Am. J. Physiol. Heart Circ. Physiol.* 295, H1572-H1579 (2008).
- 16 L. Formaggia, D. Lamponi, and A. Quarteroni, "One-dimensional models for blood flow in arteries", *Journal of engineering mathematics*, 47, 251-276 (2003).
- 17 J. Alastruey, K. H. Parker, J. Peiro', and S. J. Sherwin, "Lumped parameter outflow models for 1-D blood flow simulations: effect on pulse waves and parameter estimation", *Communications in Computational Physics*, 4, 317-336 (2008).
- 18 V. Milisic, and A. Quarteroni, "Analysis of lumped parameter models for blood flow simulations and their relation with 1D models", *ESAIM: Mathematical modelling and numerical analysis*, 38, 613-632 (2004).
- 19 Y. Shi, P. Lawford, and R. Hose, "Review of zero-D and 1-D models of blood flow in the cardiovascular system", *Biomedical engineering online*, 10, 33 (2011).
- 20 I. Kokalari, T. Karaja, and M. Guerrisi, "Review on lumped parameter method for modeling the blood flow in systemic arteries", *Journal of biomedical science and engineering*, 6, 92 (2013).
- 21 A. Quarteroni, A. Veneziani, and C. Vergara, "Geometric multiscale modeling of the cardiovascular system, between theory and practice", *Computer Methods in Applied Mechanics and Engineering*, 302, 193-252 (2016).

Having just two different morphologies (figures 2, 3, 5 etc and in the corresponding text) seems inadequate to determine a general trend or hard conclusions.

We respectfully disagree with the reviewer in this point. The main result of our manuscript (which we hope it is more clear in this corrected version) is that resonances can appear as the result of the cooperation between the network elasticity and the presence of bifurcations. We have analytically proven it for a single bifurcation in section H: origin of resonances and scaling behaviour. This constitutes a new concept in the literature, and, to our knowledge, and having reviewed the references suggested by the referee, it is the first time that it has been reported. We are sure that the referee will agree, that having shown that this effect exists already for a system as simple as a single bifurcation, it will exist for general symmetric tree-like networks, as we verify it. We have made an effort to clarify this point in the present version of the manuscript.

Much material is simply repeated in the abstract, at the end of the Introduction and in the Conclusions. Also, generally, the presentation is far too lengthy given what has been done in previous research.

We are aware that presenting the model of elastic networks makes the manuscript longer, however, since the general model was done for a general network and used to describe the arterial system, and here it is particularized for symmetric tree-like networks (for which a scalar response function can be written), we thought that it would be confusing to refer the reader to a paper whose aim was addressing the arterial system. That is why we decided to leave the model description for completeness of the presentation.

We have made an effort to reduce the length of the paper and to have different wording in Abstract, end of Introduction and Conclusions. We have largely reduced the information at the end of the Introduction.

A line of slope $\frac{1}{2}$ provides a promising comparison in figure 4 and in the accompanying text. Yet the obvious question of why $\frac{1}{2}$ is the appropriate slope is not addressed in the paper. Similarly the resonance shift to high frequencies remains not only unexplained but hardly considered.

We thank the referee for pointing this out. In order to understand the scaling behaviour in Fig 4, we notice that for a single bifurcation the GRF has terms in $K_i \kappa_i$, which represent slowly varying modes of frequency, and terms in $\cos(\kappa_i l_i)$ and $\sin(\kappa_i l_i)$ which are rapid modes, that will determine the GRF extremes, and therefore the resonances.

From the expressions for κ , C and K , and since κl is a non-dimensional quantity, we can obtain a characteristic frequency of each vessel in the system, given by $\omega_i = (E_i h_i / \rho r_i)^{1/2} / l_i$. As in many elastic systems, for instance, a forced harmonic oscillator, resonances appear when the forcing is made at the smallest characteristic frequency of the system. In this case, the frequency characteristic of the largest vessel in the network. This also explains why resonances shift to high frequencies as the system becomes more rigid (with higher values of Young moduli). We have added this paragraph in section H.

The presentation seems disorganised and inconsistent in parts.

We have made many corrections to the manuscript to have a clearer presentation.

There are conjectures but no serious links with experiments. Fluid parameters are mentioned in the section heading on page 9 but appear to be absent in that section.

To our knowledge, there are no experiments in the literature with the conditions assumed in our model that would allow a comparison. Our conjectures are presented in order to motivate future experiments.

We have changed the name of the section to: E. Network morphologies. In that one, we refer the reader to Tables II, and I for values of parameters. In particular, the fluid parameters are given in caption of Table I.

There are numerous errors of grammar scattered through the paper.

We have reviewed the English and corrected the grammar.

Reviewer #3:

"In my opinion, the paper is correct, well explained, and with interesting results. The physical discussion of the results is clear and well stated.

This work analyzes the dynamic response of an elastic tube network saturated with a viscous fluid. The main results indicate that this dynamic response of the network differs from the dynamic

response of a single tube. They found that the previous decreasing monotonic behavior of the resonant frequency is no longer always valid. In terms of the impact, I can see possible applications in terms of biophysical applications and microdevices with fluids inside.

The authors analyze, from the analytical point of view, through mathematical modeling of momentum transport equations, the problem of the dynamic response of a single tube or tube networks. The model is well constructed, and the assumptions have been established.

In previous work of the authors, they found allometric relations in terms of the longer diameter of the initial (wider) tube of the networks. The authors may give a discussion of the possible similar results in this case, where the elastic property of the tube is crucial.”

We thank the reviewer for his/her general opinion of our paper and in particular for this observation. We have added a note, regarding a characteristic frequency of the system that goes like $\omega_i = (E_i h_i / \rho r_i)^{1/2} / l_i$. As in many elastic systems, for instance, a forced harmonic oscillator, resonances appear when the forcing is made at the smallest characteristic frequency of the system. We have added an explanation of how we obtain this characteristic frequency in section H. We have limited ourselves in the paper to write this result, but it is worth it to notice, that, depending on how the combination of parameters are varied, h could be defined as a percentage of r, and E could also depend on r. We have numerically checked, that keeping constant E and constant h, and only varying the networks radii, the resonance decreases as $\omega_r \sim (1/r)^{1/2}$.

In the next figure one can see the response functions for the dog’s network when the radii of the vessels in the whole network are all multiplied by the same value

And in the next image the plot of the log of the resonance frequency (for the two first resonant peaks) as a function of the log of the factor by which the radii are multiplied. We observe that the slope of the lines is approximately -0.5.

Appendix C

Resonances in the response of fluidic networks inherent to **the cooperation between elasticity and bifurcations**

D. Yañez,¹ Rui D.M. Travasso,² and E. Corvera Poiré^{1, 2, 3, a)}

¹⁾*Departamento de Física y Química Teórica, Facultad de Química, Universidad Nacional Autónoma de México, Ciudad de México 04510, Mexico.*

²⁾*CFisUC, Department of Physics, University of Coimbra, Rua Larga, 3004-516 Coimbra, Portugal.*

³⁾*Imaging Sciences and Biomedical Engineering Division, King's College, St Thomas' Hospital, London, UK.*

(Dated: 9 August 2019)

Abstract

A global response function (GRF) of an elastic network is introduced as a generalisation of the response function (RF) of a rigid network, relating the average flow along the network with the pressure difference at its extremes. The GRF can be used to explore the frequency behaviour of a fluid confined in a tree-like symmetric elastic network in which vessels bifurcate into identical vessels. We study such dynamic response for elastic vessel networks containing viscous fluids. We find that the bifurcation structure, inherent to tree-like networks, qualitatively changes the dynamic response of a single elastic vessel, and gives resonances at certain frequencies. This implies that the average flow throughout the network could be enhanced if the pulsatile forcing at the network's inlet were imposed at the resonant frequencies. The resonant behaviour comes from the cooperation between the bifurcation structure and the elasticity of the network, since the GRF has no resonances neither for a single elastic vessel nor for a rigid network. We have found that resonances shift to high frequencies as the system becomes more rigid. We have studied two different symmetric tree-like network morphologies and found that, while many features are independent of network morphology, particular details of the response are morphology dependent. Our results could have applications to some biophysical networks, for which the morphology could be approximated to a tree-like symmetric structure and a constant pressure at the outlet. The GRF for these networks is a characteristic of the system fluid-network, being independent of the dynamic flow (or pressure) at the network's inlet. It might therefore represent a good quantity to differentiate healthy vasculatures from those with a medical condition. Our results could also be experimentally relevant in the design of networks engraved in microdevices, since the limit of the rigid case is almost impossible to attain with the materials used in microfluidics and the condition of constant pressure at the outlet is often given by the atmospheric pressure.

^{a)}Electronic mail: eugenia.corvera@gmail.com

A. Introduction

The presence of branched fluidic networks in nature is ubiquitous¹. In plants, conducts called xylem transport the required water and nutrients from the roots to stems and leaves. These vessels present a space-filling branched structure capable of reaching all cells in the organism². In mammals, blood flow carries oxygen from lungs and nutrients from the digestive system to all cells in the organism through vessel networks. Not surprisingly, several pathological scenarios associated with high mortality in humans are related to alterations in vascular networks that lead to changes in blood flow. When infarctions occur for example, vessel blockage stops blood irrigation to a tissue, leading to cell death. In the case of myocardial infarction, or heart attack, the lack of irrigation might lead to the impossibility of the heart to continue functioning³. In the case of cerebral infarction, if the brain cells lack oxygen for more than a few minutes they will die⁴. In aneurysm formation, deformation of vessel walls alter the flow profile and the stresses exerted by blood on the vessels, leading to vessel rupture^{5,6}. Hypertension is mostly associated with vessel network rigidity⁷⁻⁹. Due to the importance of blood flow to human health, understanding how the characteristics of vessel networks lead to changes of the overall network flow is of the utmost importance. In particular, blood flow in the cardiovascular system has been modeled in great detail and from different perspectives¹⁰⁻²¹. Blood flow models can have different levels of detail depending on the phenomena they aim at probing. Complex 3D Navier-Stokes calculations coupled with a detailed elastic treatment of vessel dynamics are able to model the flow in a blood cycle at every point within a vessel, however their level of complexity make them prohibitive when studying blood flow in a large vascular network. In these situations 0D or 1D models are convenient^{13,17-20}.

Several microfluidic devices are also constructed as branched network of vessels, often with the aim of delivering fluid to a large number of sub-devices^{22,23}. A characterisation of how material properties, like elasticity, influence the global response function (GRF) in these branched microfluidic networks, may permit to better tailor the network morphologies and materials according to the need²⁴.

The response function (RF) of tree-like symmetric rigid networks, in which vessels bifurcate into identical vessels, relates the flow that goes through the network, with the pressure

difference between the network extremes²⁵. It is a concept extrapolated from the dynamic permeability of fluids confined in circular or rectangular channels^{26–33}, and it is essentially an effective permeability times and effective area²⁵. Knowledge of the dynamic permeability in **rigid** channels, or of the response function in **rigid** networks, is important since it allows one to know a-priori the frequencies of the pressure signal that enhance the flow^{30,31,33–36}.

Several previous works of our group have shown that the RF depends strongly on the morphological properties of the network, on the rheological properties of the fluid and on the frequencies involved in the pressure pulse^{34–36}. These works have been done for rigid vessel networks.

Recently, it has been found that the **RF** of a Newtonian fluid flowing in a single elastic vessel, **that is able to deform along the flow direction**, might have striking effects as a function of frequency in elastomeric materials at microscales³⁷. A **GRF for a tree-like symmetric elastic network**, introduced as a generalisation of the RF of a rigid network, relating the average flow along the network with the pressure difference at its extremes, has been introduced in^{38,39}, but its behaviour as a function of frequency is yet to be studied.

In this paper we study the **GFR** of tree-like symmetric elastic vessel networks, and explore the effect that the degree of elasticity and network morphology have on this frequency-dependent response function. Our study is relevant in microfluidic devices, where for a given pressure drop, flow rate in a deforming channel is found to be several times higher than the one expected in a non-deforming channel⁴⁰. It could also be relevant for **physiological** vessel networks, since **they are formed by** elastic structures.

In sections B and C, for thoroughness of the presentation, we briefly describe a model for flow in elastic networks that has been published and validated for the arterial network in⁴¹. In section B, we present the basic considerations to study flow in a single elastic vessel. In section C, we state the necessary considerations to apply the model to elastic vessel networks. In section D, we introduce the global response function for tree-like symmetric elastic networks. In section E, we describe, the two tree-like network morphologies that are used in this work. In section F, we find that the GRF is independent of the dynamics of the inflow, for networks that have constant pressure at the outlets, making the GRF a good quantity to study the network's dynamics. In section G, we show that the bifurcation

structure of tree-like elastic networks causes the GRF to have resonances, which do not exist for rigid networks, nor for single elastic vessels. This implies, that the flow magnitude across the network could be enhanced at certain frequencies due to the cooperation between the bifurcation structure and the elasticity of the network, via pulsatile forcing. We do a systematic study varying the networks' elasticity, and find features that are common to different network morphologies, and features that are morphology dependent. In section H, we present an analytical study of a single elastic bifurcation that demonstrates the emergence of the resonant behaviour. We present our conclusions in section I.

B. Flow through a single elastic vessel

To study the dynamics of each elastic vessel forming the network, we use the Generalised Darcy's Elastic Model (GDEM)⁴¹, which assumes that:

1.- A generalised Darcy's model for rigid cylindrical vessels, which gives a linear relation between flow, \hat{q} , and pressure gradient, $\frac{\partial \hat{p}}{\partial x}$, in frequency domain, is valid locally, that is, at any point x along the flow direction,

$$\hat{q} = -\frac{AK(\omega)}{\eta} \frac{\partial \hat{p}}{\partial x}, \quad (1)$$

where $K(\omega) = -\frac{\eta}{i\omega\rho} \left[1 - \frac{2J_1(\beta r)}{\beta r J_0(\beta r)}\right]$ stands for the dynamic permeability of a Newtonian fluid, which is a measurement of the resistance to flow; J_0 and J_1 are Bessel functions of the first kind of orders 0 and 1 respectively; $\beta^2 = \frac{i\rho\omega}{\eta}$, r is the average radius of the elastic vessel, η is the fluid viscosity, and $i = \sqrt{-1}$. Hats over quantities denote their Fourier transforms. Flow has been approximated as a velocity averaged over the average cross sectional area, A , times this area.

2.- A linear relationship between pressure and flow gradient, coming from mass conservation and a Hooke-like linear model for vessel elasticity, is valid locally, namely

$$\hat{p} = \frac{1}{i\omega C} \frac{\partial \hat{q}}{\partial x}, \quad (2)$$

where $C = \frac{3\pi r^3}{2Eh}$ is called the vessel compliance, h is the vessel wall thickness, E is the Young modulus of the vessel given by

$$E = 3\rho c^2 r / (2h), \quad (3)$$

where c is the pulse wave velocity. The limit of a rigid vessel is obtained when the vessel compliance vanishes, that is, when $C \rightarrow 0$.

Combining Eqs. 1 and 2 one obtains a harmonic oscillator equation for the pressure in frequency domain, namely,

$$\frac{\partial^2 \hat{p}}{\partial x^2} = -\kappa^2 \hat{p}, \quad (4)$$

where $\kappa^2 = \frac{i\omega C\eta}{AK(\omega)}$.

Solving Eq. 4 for a single vessel, of length l , with known \hat{p}_{in} and \hat{p}_{out} for the inlet and outlet pressures on the vessel extremes, respectively, yields an analytical expression for the pressure along the vessel in frequency domain given by

$$\hat{p}(x) = \hat{p}_{in} \cos(\kappa x) + \frac{\hat{p}_{out} - \hat{p}_{in} \cos(\kappa l)}{\sin(\kappa l)} \sin(\kappa x), \quad (5)$$

which, when substituted in Eq. 1, provides the flow along the vessel in frequency domain,

$$\hat{q}(x) = M \left(\hat{p}_{in} \sin(\kappa x) - \frac{\hat{p}_{out} - \hat{p}_{in} \cos(\kappa l)}{\sin(\kappa l)} \cos(\kappa x) \right), \quad (6)$$

where $M^2 = i\omega C A K(\omega) / \eta$, and $x \in [0, l]$.

Solving Eq. 4 for a single vessel with known \hat{q}_{in} and \hat{p}_{out} for the inlet flow and outlet pressure on the vessel extremes, respectively, yields an analytic expression for the pressure along the vessel in frequency domain given by

$$\hat{p}(x) = \frac{\hat{q}_{in} \sin(\kappa l) + M \hat{p}_{out}}{M \cos(\kappa l)} \cos(\kappa x) - \frac{\hat{q}_{in}}{M} \sin(\kappa x), \quad (7)$$

which, when substituted in Eq. 1, provides the flow along the vessel in frequency domain,

$$\hat{q}(x) = \frac{\hat{q}_{in} \sin(\kappa l) + M \hat{p}_{out}}{\cos(\kappa l)} \sin(\kappa x) + \hat{q}_{in} \cos(\kappa x). \quad (8)$$

C. Flow through an elastic vessel network

GDEM⁴¹ considers elastic networks of tubes in which a viscous fluid flows subject to a time dependent inlet flow (or pressure). In this paper, we focus on tree-like symmetric networks, in which cylindrical vessels bifurcate into two identical vessels, as illustrated in Fig. 1 for a 4-level network. The model deliberately excludes trifurcations or higher order branching,

FIG. 1. Illustration of a 4-level bifurcating tree-like network.

since bifurcations are the most probable and common forms of branching structures in nature. Articles in the literature, that classify normal vascular networks, consider only bifurcations^{42–44}. More complex branching morphologies, such as trifurcations, are present in non-hierarchical pathological vasculatures, such as the ones irrigating tumours⁴⁵.

Tree-like networks will have two types of vessels: the ones in which boundary conditions are imposed, namely vessels that take inlet flows or pressures as boundary condition, or vessels in which outlet pressures are imposed; and vessels that are internal to the network. For these ones, the pressures at the vessels extremes have to be determined as part of the solution.

Flow conservation is imposed at the nodes (points of bifurcation) as well as equal pressures on the extremes of the vessels connected to a node. For instance, for a vessel at level j , with length l_j , that bifurcates into vessels at level $j+1$, with lengths l_{j+1} , flow conservation at the node joining these three vessels, requires the output flow at level j , $\hat{q}_j(x = l_j)$ to be equal to twice the input flow at level $j+1$, $\hat{q}_{j+1}(x = 0)$, that is, $\hat{q}_j(x = l_j) = 2\hat{q}_{j+1}(x = 0)$. This allows one to write an equation, by means of Eq. 6 (or Eq. 8), that involves: pressure (or flow) at the entrance of vessel at level j , pressure at the node, and pressures at the outlet ($x = l_{j+1}$) of vessels at level $j+1$. For each node in the network, there will be an equation coming from flow conservation at that node. The result is a system of equations for the pressure at the nodes, that can be solved, analytically or numerically, using the methodology described

in ⁴¹.

Once the pressure at the nodes is known, pressure and flow as a function of the position, x , along any vessel of the network, can be obtained by making use of Eqs. 5, 6, 7 and 8.

D. Global response for a tree-like symmetric elastic network

For a rigid network, the fact that flow is constant along the network, allows one to describe the network behaviour in terms of a dynamic response function, $\chi(\omega)$, which is a measure of how much flow, $q_1(t)$, goes through the network, given a pressure difference, $\Delta p(t) = p_{out} - p_{in}$, between the network inlet and the network outlet²⁵. In frequency domain, this can be written as

$$\hat{q}_1 = -\frac{\chi(\omega)\Delta\hat{p}(\omega)}{\eta L}, \quad (9)$$

where $L = \sum_i l_i$, is the sum of the vessels' length, l_i , at each level i . χ has been computed for a tree-like symmetric rigid network by means of an electrical analogy, adding resistances $\frac{l_i}{A_i K_i}$ in series or parallel, according to the network morphology²⁵, giving the following result

$$\frac{1}{\chi} = \frac{1}{L} \sum_{i=1}^N \frac{l_i}{2^{i-1} A_i K_i}. \quad (10)$$

In Eq. 9, q_1 is the flow at the inlet of the network, but given that flow is constant along a rigid network, it is also the total flow at any network level, i , containing 2^{i-1} vessels, that is, $q_1 = 2^{i-1} q_i$, where q_i is the flow in each of the vessels at level i .

For an elastic network, on the other hand, flow is not uniform along the network. Hence, we define a global response function (GRF), that would give information of the flow, integrated throughout the network, that is, the flow averaged along the flow direction, $\langle Q \rangle_x(t)$. A global response function for a tree-like symmetric elastic network is defined, in frequency domain, as in^{38,39}

$$\chi_{global} \equiv -\frac{\eta L \langle \hat{Q} \rangle_x}{\Delta\hat{p}}, \quad (11)$$

where

$$\langle \hat{Q} \rangle_x = \frac{1}{L} \sum_{i=1}^N \int_0^{l_i} 2^{i-1} \hat{q}_i(x) dx = \frac{1}{L} \sum_{i=1}^N 2^{i-1} l_i \langle \hat{q}_i \rangle_x. \quad (12)$$

Here $\hat{q}_i(x)$ is the flow at position x in one of the vessels at level i ; $2^{i-1}\hat{q}_i(x)$ is the total flow at position x in level i , and $\langle \hat{q}_i \rangle_x = \frac{1}{l_i} \int_0^{l_i} \hat{q}_i(x) dx = -\frac{A_i K_i}{\eta l_i} \Delta \hat{p}_i$, is the average flow along a single vessel at level i . The second equality in this expression is obtained using Eq. 1. $\Delta \hat{p} = \sum_{i=1}^N \Delta \hat{p}_i$. Note that for a rigid network, $\langle \hat{Q} \rangle_x$ reduces to \hat{q}_1 , which is the flow at point x , of the network's cross sectional area. In the limit of a rigid network, Eq. 11 reduces to Eq. 9, with $\chi_{global} = \chi$, given by Eq. 10.

E. Network morphologies

In order to consider the impact that network morphology has on the dynamics, we study two different tree-like symmetric networks.

One of them, already used in previous works^{25,34}, approximates the morphological properties of the vasculature of a dog. The vessel morphology of the dog's circulatory system is described in detail in the literature⁴⁷, and we have approximated the number of vessels, lengths and radii to the closest numbers that satisfy branching into identical vessels (see Table I). For convenience in the discussion, we will refer to this network as the “dog's network”.

We also analyse networks whose vessels follow Murray's law⁴⁸, since physiological studies have validated the agreement of Murray's law in vascular systems of mammals⁴⁹ and even in networks that transport water in plants⁵⁰. According to Murray's law, when a vessel of radius r_p bifurcates into two identical vessels, the next level vessels are narrower, with radii r_d , that obey $r_p^3 = 2r_d^3$. Therefore, the radii of vessels at level i , are related to the inlet vessel radius, r_1 , by

$$r_i = \left(\frac{1}{2}\right)^{(i-1)/3} r_1. \quad (13)$$

For a Murray's network starting with the dimensions of the dog's aorta and containing the same number of levels as the “dog's network”, the radii of all levels is prescribed by Eq. 13. We have adjusted a power law for the lengths, as in³⁵. Parameters used in the calculations are reported in Table II. The caption of Table I includes the fluid parameters used in this work.

F. Independence on Boundary conditions

For a network response like the one in Eq. 11 to be useful, it must prove to be independent on boundary conditions. We limit the study to the case of networks whose pressure is constant at the outlet, and consider three different inflows as incident boundary condition. Namely, an aortic physiological flow measured in-vivo taken from⁵², a simple time-dependent one-mode cosine function, and a simple pulse consisting of a delta function at time zero. We have chosen a constant pressure at the network outlet of 4.2 kPa for the three calculations, a value that is consistent with physiological situations. Following the methodology sketched out in section C, and explained in detailed in reference⁴¹, we have computed flows and pressures all along the network. We have used them to compute the response function defined in Eq. 11. We have verified that the response functions of the network, using these three different inflows, and constant pressure at the network outlet, are identical. Since, by construction, the inflow consisting of a simple pulse in time domain, is constant in frequency domain, it has less noise than the other signals and it is the one reported in all figures. We have also verified that halving or doubling the inflows or the outlet pressures do not alter the response function.

G. Results

Fig. 2 shows the absolute value of the response function for two different networks. The first thing to notice is that both responses are non-monotonic functions of frequency and therefore present resonances (maxima at certain frequencies). This implies that the flow magnitude would be enhanced at certain frequencies of the pulsatile forcing. This enhancement comes from the cooperation between the bifurcation structure and the elasticity of the network via pulsatile forcing. Elasticity by itself does not cause resonances in the flow of Newtonian fluids subject to pulsatile forcing. That is, for a single vessel, the response function, χ_{global} , given by Eq. 11, decreases monotonically with frequency (this will be explicitly shown in section H). The structure of bifurcations is necessary to have resonances.

There is an interplay between network geometry and network elasticity to give the exact location of resonances. In Fig. 2, we can observe that the main resonance (the one with the

FIG. 2. Magnitude of the response function as a function of frequency for the dog’s and Murray’s elastic networks for “physiological” Young moduli, E_0 . Parameters for the calculation are given in Tables I and II.

highest value of the response function) for Murray’s network occurs around $\omega = 2$ rad/s, while for the dog’s network occurs around $\omega = 4$ rad/s. Values of the response function at resonance are $4 \times 10^{-11} \text{ m}^4$ and $5 \times 10^{-11} \text{ m}^4$, respectively. These values are surprisingly close to each other, given that the steady state network responses (at low frequencies, out of the range of Fig. 2) are five orders of magnitude different. This is because the outer, wider vessels, which are of a similar caliber in both networks, determine the high frequency behaviour of the response. The elastic properties of wider vessels allow for better accumulation and release of fluid during oscillatory flow, causing the non-monotonic high frequency behavior of the response function.

The presence of vessel elasticity changes qualitatively the dynamic response of rigid networks, for which there are no resonances. Fig. 3 shows, how the response function changes as the Young moduli of the networks, E , are varied as multiples of E_0 . For both networks Murray’s in Fig. 3a and the dog’s one in Fig. 3b, resonances shift to high frequencies as the system becomes more rigid. For both networks, resonance frequencies are proportional

FIG. 3. a) Magnitude of the response function as a function of frequency for Murray's elastic networks with various degrees of elasticity. b) Magnitude of the response function as a function of frequency for the dog's elastic network with various degrees of elasticity. In both figures the magnitude of the response of the corresponding rigid network is shown in black.

to the square root of the network's Young moduli, that is, $\omega_{res} \sim (\frac{E}{E_0})^{1/2}$ (see section H). Resonance frequencies as a function of network's elasticity are shown in red in Fig. 4a) for

Murray’s network. The continuous line, shown for reference, has a slope of $1/2$. For the dog’s network, frequencies for the first two maxima are shown in Fig. 4b). In between the points indicating the frequencies of the first and second maxima, there is a continuous line, shown for reference, with a slope of $1/2$. As the network becomes more rigid, the first maximum takes over and becomes the main resonance of the system. The main resonance is plotted in red.

We have also found that the magnitude of the responses at resonance decreases for increasing network rigidity. This can be appreciated in Fig. 3a). Resonances disappear, for large network rigidities, when the values of the response function, at all frequencies, are below the value of the response at zero frequency (not shown). These are features of the response function that are independent of the network morphologies studied. Particular details of the response function are morphology dependent. For Murray’s network (Fig. 3a), there is a frequency region around the first resonance where the value of the response is larger than for a rigid network, while the high frequency behaviour presents responses below the one of the rigid network and the low frequency behavior is quite similar to the one of the rigid network. In contrast, for the dog’s network (Fig. 3b), vessel elasticity changes dramatically the low frequency behaviour of the response, causing the network response to increase as a function of frequency. This is because inner, thinner vessels (with high resistance and small permeability), which are much smaller for the dog’s network than for the Murray’s network, determine the low frequency behaviour of the response function. To accommodate the low frequency behaviour of the response (given by the small vessels, with large resistance and low permeability) and the high frequency behaviour of the response (given by the outer wider vessels, with low resistance and high permeability), the network’s response increases as a function of frequency. This behaviour is qualitatively different from the monotonic decrease of the response function for rigid networks. Also, for the dog’s network, the value of the response increases by several orders of magnitude, for most of the frequency spectrum, relative to the one of a rigid network. This implies that the maximum amplitudes of the overall flow, regardless of the frequencies involved in the pressure drop across the network, would increase significantly with respect to the rigid case.

FIG. 4. a) Resonance frequency for a Murray's network as a function of network's elasticity. b) Frequencies for the first two maxima of the dog's network as a function of network's elasticity. As the network becomes more rigid, the first maximum takes over and becomes the dominant resonance of the system. The dominant resonance is plotted in red. Continuous black lines in both figures, have a slope of $1/2$, and are shown for reference.

H. Origin of resonances and scaling behaviour

As stated in section G, we have found resonance frequencies for the GRF of a Newtonian fluid in an elastic network, using a model that for a Newtonian fluid in a single elastic vessel does not exhibit resonances. This implies that the resonant behaviour is due to the structure of bifurcations, inherent to tree-like networks. In order to better see this, we compute analytically the GRF for a network consisting of a single bifurcation, and show the emergence of the non-monotonic behavior as a function of frequency.

The response for a Newtonian fluid in a single elastic vessel, with average cross sectional area, A , can be obtained analytically from the equation for flow along the vessel (Eq. 6), that, when integrated along the flow direction, gives:

$$\langle \hat{q} \rangle_x = -\frac{AK(\omega)}{\eta l}(\hat{p}_{out} - \hat{p}_{in}), \quad (14)$$

which gives a response function, χ_{global} , defined in Eq. 11, that is equal to the area, A , times the dynamic permeability, $K(\omega)$, *i.e.*, $\chi_{global} = AK(\omega)$, with $K(\omega)$ given by the analytical expression after Eq. 1 and, for a Newtonian fluid, has a monotonic decay as a function of frequency^{26,31}.

We now consider a network with a single bifurcation and compute its response function. We consider a vessel at level 1, that bifurcates into two identical vessels at level 2. Flow conservation at the bifurcation implies that outflow in vessel at level 1, is equal to the sum of inflows of vessels after the bifurcation, namely $\hat{q}_1(x = l_1) = 2\hat{q}_2(x = 0)$. For vessel at level 1, the pressure at the vessel's entrance, is the pressure at the entrance to the system, \hat{p}_{in} , and the pressure at the exit is an unknown pressure at the node, \hat{p}_N . For vessels at level 2, the pressure at the entrance is the pressure at the node, \hat{p}_N , and the pressure at the vessels' exit is the output pressure of the system, \hat{p}_{out} . Using Eq. 6, in the equation for flow conservation at the node, we can write an equation for the pressure at the node, \hat{p}_N . This one is given by

$$\hat{p}_N = \frac{2A_2K_2\kappa_2 \sin(\kappa_1 l_1)\hat{p}_{out} + A_1K_1\kappa_1 \sin(\kappa_2 l_2)\hat{p}_{in}}{A_1K_1\kappa_1 \sin(\kappa_2 l_2)\cos(\kappa_1 l_1) + 2A_2K_2\kappa_2 \sin(\kappa_1 l_1)\cos(\kappa_2 l_2)} \quad (15)$$

On the other hand (using Eq. 12), the average flow along the network with a single bifurcation

FIG. 5. Magnitude of the response function for a single bifurcation.

is

$$\langle \hat{Q} \rangle_x = \frac{1}{l_1 + l_2} [l_1 \langle \hat{q}_1 \rangle_x + 2l_2 \langle \hat{q}_2 \rangle_x], \quad (16)$$

which, using Eq. 14, can be written as

$$\langle \hat{Q} \rangle_x = -\frac{[2A_2K_2\hat{p}_{out} - A_1K_1\hat{p}_{in}]}{\eta(l_1 + l_2)} - \frac{(A_1K_1 - 2A_2K_2)}{\eta(l_1 + l_2)}\hat{p}_N, \quad (17)$$

with \hat{p}_N given by Eq. 15. Accordingly, the response function, defined in Eq. 11, is given by

$$\chi_{global} = \frac{[2A_2K_2\hat{p}_{out} - A_1K_1\hat{p}_{in}] + (A_1K_1 - 2A_2K_2)\hat{p}_N}{\hat{p}_{out} - \hat{p}_{in}}, \quad (18)$$

From this expression, it becomes clear why there is a non-monotonic behaviour as a function of frequency coming from the bifurcation. That is, the pressure at the node, \hat{p}_N , needed to compute χ_{global} , and given by Eq. 15, contains non-monotonic sinusoidal terms in $\kappa_1 l_1$ and $\kappa_2 l_2$, that are functions of frequency and of the mechanical properties of the vessels (see expression for κ after Eq. 4). These terms were not averaged out when the integration along the flow was performed, as it happened for a single vessel.

Fig. 5 illustrates the response function for a network, that consist of a single bifurcation, with the characteristics of the first three vessels for both, the dog's and Murray's network.

We have chosen a zero pressure at the outlet. The results clearly show the non-monotonic behaviour coming from the bifurcation. For this particular example

$$\chi_{global} = A_1 K_1 - (A_1 K_1 - 2A_2 K_2) \frac{\hat{p}_N^*}{\hat{p}_{in}}, \quad (19)$$

where

$$\frac{\hat{p}_N^*}{\hat{p}_{in}} = \frac{A_1 K_1 \kappa_1 \sin(\kappa_2 l_2)}{A_1 K_1 \kappa_1 \sin(\kappa_2 l_2) \cos(\kappa_1 l_1) + 2A_2 K_2 \kappa_2 \sin(\kappa_1 l_1) \cos(\kappa_2 l_2)}. \quad (20)$$

For this example, the response function, χ_{global} , is explicitly independent from the pressure at the inlet. It is straightforward to prove, that for constant $p_{out}(t)$, χ_{global} is independent of the pressure drop.

In order to understand the scaling behaviour observed in Fig 4, we notice that the GRF has terms in $K_i \kappa_i$, which represent slowly varying modes of frequency, and terms in $\cos(\kappa_i l_i)$ and $\sin(\kappa_i l_i)$, which are rapid modes, that will determine the GRF extremes, and therefore the resonances. From the expressions for K , C and κ (after equations Eq. 1, Eq. 2 and Eq. 4), and since κl is a non-dimensional quantity, we can obtain a characteristic frequency of each vessel in the system, given by $\omega_i = \frac{1}{l_i} \left(\frac{E_i h_i}{\rho r_i} \right)^{1/2}$. As in many elastic systems, for instance, a forced harmonic oscillator, resonances appear when the forcing is made at the smallest characteristic frequency of the system. In this case, the frequency characteristic of the largest vessel in the network. This also explains why resonances shift to high frequencies as the system becomes more rigid (with higher values of Young moduli).

I. Conclusions

A global response function (GRF) of a tree-like symmetric elastic network, is introduced as a generalisation of the response function of a rigid network^{38,39}. The (GRF) relates the network's flow, averaged along the flow direction, with the pressure difference at the network's extremes. It can be used to explore the frequency behaviour of a fluid confined in an elastic network. The GRF indicates which frequencies, involved in the dynamic pressure drop, maximize the magnitude of flow averaged along the flow direction. We have found resonance frequencies of the GRF for Newtonian fluids in elastic networks using a model that for a single elastic vessel, and for rigid networks, does not give resonances, and proved that this resonant behavior is due to the cooperation between elasticity and bifurcations.

Some of the features of the GRF are common to networks of different morphologies, for instance, for all networks, resonance frequencies shift to high frequencies as the system becomes more rigid. For all of them, responses at resonance decrease for increasing network rigidity. Particular details of the response function are morphology dependent. For example, in the dog's network studied here, vessel elasticity changes dramatically the low frequency behaviour of the GRF, causing this one to increase as a function of frequency. This behaviour could be experimentally important for certain networks engraved in microdevices, since the limit of the rigid case is almost impossible to attain with the materials used in microfluidics.

For network's in which pressure is constant at the outlets, the GRF is characteristic of the system fluid-network, and independent of the dynamics of the inflow and of the value of pressure at the network's outlet. It might therefore represent a good quantity to differentiate healthy vasculatures from those with a medical condition. Abnormalities in large vessels could possibly be observed in the high frequency behaviour of the GRF, while abnormalities in small vessels would in principle be observable in the low frequency behaviour of the GRF. Whether or not this quantity might be clinically relevant to discriminate vasculatures with a medical condition, from those of a control group, is yet to be explored.

Our methodology could also be applicable to the domain of microfluidics where, branched symmetric structures are often engraved in microchips whose materials range from elastomeric to rigid. For a possible experimental verification of our results, it would be worth recalling that, for given values of the elastic and fluid parameters of a microfluidic device, one can always attain the linear flow regime by decreasing the amplitude of the dynamic pressure drop.

Data accessibility

Not Applicable. The results presented in the figures are the direct calculations of the respective equations for the response functions.

Competing Interests

The authors declare no competing interests.

Authors' contributions:

DY, RT, EC performed the analytical and numerical calculations. RT, EC wrote the article. EC coordinated the work. All authors gave final approval for publication.

Funding

DY and ECP thank funding from CONACyT (Mexico), through project 219584, and the Faculty of Chemistry UNAM, through *subprograma* 127. RDMT thanks the support of FEDER funds through the Operational Program Competitiveness Factors - COMPETE and to national funds by FCT - Foundation for Science and Technology under the strategic project UID/FIS/04564/2016 and under POCI-01-0145-FEDER-031743 - PTDC/BIA-CEL/31743/2017. ECP thanks funding from CONACyT (Mexico) through agreement 2018-000007-01EXTV-00183; and from DGAPA, UNAM through a PASPA program, during a sabbatical leave.

Research ethics

We were not required to complete an ethical assessment prior to conducting our research.

Animal ethics

Not applicable.

Permission to carry out fieldwork

Not applicable.

I. TABLES OF PARAMETERS USED IN THE CALCULATIONS

Levels	# of vessels	Radius (μm)	Length (cm)	E_0 (MPa)
1	1	5000	40.0	0.70
2-5	30	1500	20.0	1.4
6-9	480	500	10.0	2.8
10-11	1536	300	1.0	3.8
12-25	33552354	10	0.2	30
26-29	503316480	4	0.1	50

TABLE I. Number and characteristics of vessels for the different levels of the dog's network. Taken from²⁵, and based on the anatomical measurements collected in⁴⁷. Typical dimensions of vessel 1 are those of the aorta, typical dimensions of vessels 2 – 5 are those of large arteries; of vessels 6 – 9 are those of main arterial branches, of vessels 10 – 11 are those of terminal branches, of vessels 12 – 25 are those of arterioles and of vessels 26 – 29 are those of capillaries. The vessel wall width, h , was taken to be equal to $h = 0.1r$, where r is the radius. Values used for the fluid viscosity and density were $\eta = 5.0 \times 10^{-3}$ kg/(m·s) and $\rho = 1050$ kg/m³, respectively⁵¹. The Young modulus E is given by Eq.3. For an arterial tree, the pulse wave velocity, c , is given by the empirical relationship $c = 13.3/(2r)^{0.3}$ (in m/s), with r measured in mm⁴⁶. This gives the values of E_0 in the table.

Levels	Radius (μm)	Length (cm)	E_0 (MPa)
1	5000	40.0	0.70
2-28	$5000/2^{(n-1)/3}$	$40 n^{-1.78}$	$1.84 r^{-0.6}$
29	7.8	0.1	34

TABLE II. Number and characteristics of vessels for the different levels of Murray’s network. In this network vessel radii and lengths are obtained as a function of level n . The vessel Young’s modulus is a function of its radius r (measured in μm). The vessel wall width, h , was taken to be equal to $h = 0.1r$ in all cases. Values for fluid viscosity and density are as in Table I. The Young modulus E is given by Eq.3. For an arterial tree, the pulse wave velocity, c , is given by the empirical relationship $c = 13.3/(2r)^{0.3}$ (in m/s), with r measured in mm⁴⁶. This gives the expression for E_0 shown in the table, with r measured in mm.

REFERENCES

- ¹G. B. West, J. H. Brown, and B. J. Enquist, B. J., “A general model for the origin of allometric scaling laws in biology”, *Science* 276, 122-126 (1997).
- ²J. S. Sperry, “Evolution of water transport and xylem structure”, *International Journal of Plant Sciences* 164, S115-S127 (2003).
- ³H. D. White, and D. P. Chew, “Acute myocardial infarction”, *The Lancet* 372, 570-584 (2008).
- ⁴J. Bamford, P. Sandercock, M. Dennis, C. Warlow, and J. Burn, “Classification and natural history of clinically identifiable subtypes of cerebral infarction”, *The Lancet* 337, 1521-1526 (1991).
- ⁵A. Valencia, H. Morales, R. Rivera, E. Bravo, and M. Galvez, “Blood flow dynamics in patient-specific cerebral aneurysm models: the relationship between wall shear stress and aneurysm area index”, *Medical Engineering and Physics* 30, 329-340 (2008).
- ⁶J. R. Cebal, M. Vazquez, D. M. Sforza, G. Houzeaux, S. Tateshima, et al, “Analysis of hemodynamics and wall mechanics at sites of cerebral aneurysm rupture”, *Journal of Neurointerventional Surgery* 7, 530-536 (2015).

- ⁷R. S. Cunha, B. Pannier, A. Benetos, J. P. Siché, G. M. London, et al, “Association between high heart rate and high arterial rigidity in normotensive and hypertensive subjects”, *Journal of Hypertension* 15, 1423-1430 (1997).
- ⁸M. L. Bots, A. Hofman, A. M. de Bruyn, P. T. de Jong, and D. E. Grobbee, “Isolated systolic hypertension and vessel wall thickness of the carotid artery. The Rotterdam Elderly Study”, *Arteriosclerosis, Thrombosis, and Vascular Biology* 13, 64-69 (1993).
- ⁹S. Vennin, Y. Li, M. Willemet, H. Fok, H. Gu, et al, “Identifying Hemodynamic Determinants of Pulse Pressure. A Combined Numerical and Physiological Approach”, *Hypertension* 70, 1176-1182 (2017).
- ¹⁰F. N. Van de Vosse, “Mathematical modelling of the cardiovascular system”, *Journal of Engineering Mathematics* 47, 175-183 (2003).
- ¹¹F. T. Smith, R. Purvis, S. C. R. Dennis, M. A. Jones, N. C. Ovensden, and M. Tadjfar, “Fluid flow through various branching tubes”, *Journal of Engineering Mathematics*, 47, 277-298 (2003).
- ¹²A. R. Pries, and T. W. Secomb, “Blood flow in microvascular networks”, in *Microcirculation*, pp. 3-36, Academic Press (2008).
- ¹³M. S. Olufsen, and A. Nadim, “On deriving lumped models for blood flow and pressure in the systemic arteries”, in *Computational Fluid and Solid Mechanics*, pp. 1786-1789, Elsevier Science Ltd (2003).
- ¹⁴Y. Gandica, T. Schwarz, O. Oliveira, and R. D. Travasso, “Hypoxia in vascular networks: a complex system approach to unravel the diabetic paradox”, *PloS one*, 9, e113165 (2014).
- ¹⁵B. E. Carlson, J. C. Arciero, and T. W. Secomb, “Theoretical model of blood flow auto-regulation: roles of myogenic, shear-dependent, and metabolic responses”, *Am. J. Physiol. Heart Circ. Physiol.* 295, H1572-H1579 (2008).
- ¹⁶L. Formaggia, D. Lamponi, and A. Quarteroni, “One-dimensional models for blood flow in arteries”, *Journal of engineering mathematics*, 47, 251-276 (2003).
- ¹⁷J. Alastruey, K. H. Parker, J. Peiró, and S. J. Sherwin, “Lumped parameter outflow models for 1-D blood flow simulations: effect on pulse waves and parameter estimation”, *Communications in Computational Physics*, 4, 317-336 (2008).
- ¹⁸V. Milisic, and A. Quarteroni, “Analysis of lumped parameter models for blood flow simula-

- tions and their relation with 1D models”, *ESAIM: Mathematical modelling and numerical analysis*, 38, 613-632 (2004).
- ¹⁹Y. Shi, P. Lawford, and R. Hose, “Review of zero-D and 1-D models of blood flow in the cardiovascular system”, *Biomedical engineering online*, 10, 33 (2011).
- ²⁰I. Kokalari, T. Karaja, and M. Guerrisi, “Review on lumped parameter method for modeling the blood flow in systemic arteries”, *Journal of biomedical science and engineering*, 6, 92 (2013).
- ²¹A. Quarteroni, A. Veneziani, and C. Vergara, “Geometric multiscale modeling of the cardiovascular system, between theory and practice”, *Computer Methods in Applied Mechanics and Engineering*, 302, 193-252 (2016).
- ²²A. Olanrewaju, M. Beaugrand, M. Yafia, and D. Juncker, “Capillary microfluidics in microchannels: from microfluidic networks to capillary circuits”, *Lab on a Chip* 18, 2323-2347 (2018).
- ²³M. Hitzbleck, L. Gervais, and E. Delamarche, “Controlled release of reagents in capillary-driven microfluidics using reagent integrators”, *Lab on a Chip* 11, 2680-2685 (2011).
- ²⁴H. A. Stone, A. D. Stroock, and A. Ajdari, “Engineering flows in small devices: microfluidics toward a lab-on-a-chip”, *Annual Review of Fluid Mechanics* 36, 381-411 (2004).
- ²⁵J. Flores, E. C. Poiré, J. Del Río, and M. L. de Haro, “A plausible explanation for heart rates in mammals”, *Journal of Theoretical Biology* 265, 599-603 (2010).
- ²⁶M.-Y. Zhou, and P. Sheng, “First-principles calculations of dynamic permeability in porous media”, *Physical Review B* 39, 12027 (1989).
- ²⁷M. López de Haro, J. A. del Río, and S. Whitaker, “Flow of Maxwell Fluids in Porous Media”, *Transport Porous Med.* 25, 167 (1996).
- ²⁸J. A. del Río, M. López de Haro, and S. Whitaker, “Enhancement in the dynamic response of a viscoelastic fluid flowing in a tube”, *Phys. Rev. E* 58, 6323 (1998).
- ²⁹J. R. Castrejón Pita, J. A. del Río, A. A. Castrejón Pita and G. Huelsz, “Experimental observation of dramatic differences in the dynamic response of Newtonian and Maxwellian fluids”, *Phys. Rev. E* 68, 046301 (2003).
- ³⁰R. Collepardo-Guevara and E. Corvera Poiré, “Controlling viscoelastic flow by tuning frequency during occlusions”, *Phys. Rev. E* 76, 026301 (2007).

- ³¹M. Castro, M. E. Bravo-Gutiérrez, A. Hernández-Machado, and E. Corvera Poiré, “Dynamic Characterization of Permeabilities and Flows in Microchannels”, *Physical Review Letters* 101, 224501 (2008).
- ³²E. Corvera Poiré and A. Hernández-Machado, “Frequency-induced stratification in viscoelastic microfluidics,” *Langmuir* 26, 15084 (2010).
- ³³M. E. Bravo-Gutiérrez, M. Castro, A. Hernández-Machado, and E. Corvera Poiré, “Controlling viscoelastic flow in microchannels with slip,” *Langmuir* 27, 2075 (2011).
- ³⁴J. Flores, A. M. Romero, R. D. M. Travasso, and E. Corvera Poiré, “Flow and anastomosis in vascular networks”, *Journal of Theoretical Biology* 317, 257 (2013).
- ³⁵A. M. Torres Rojas, A. M. Romero, I. Pagonabarraga, R. D. M. Travasso, and E. Corvera Poiré, “Obstructions in Vascular Networks: Relation Between Network Morphology and Blood Supply”, *PLoS ONE* 10, e0128111 (2015).
- ³⁶A. M. Torres Rojas, R. D. M. Travasso, I. Pagonabarraga, and E. Corvera Poiré, “When do redundant fluidic networks outperform non-redundant ones?”, *Europhysics Letters* 117, 64002 (2017).
- ³⁷A. M. Torres Rojas, I. Pagonabarraga, and E. Corvera Poiré, “Resonances of Newtonian fluids in elastomeric microtubes”, *Physics of Fluids* 29, 122003 (2017).
- ³⁸L. González Mena, Treball de Fi de Màster. Universitat de Barcelona, June 2016.
- ³⁹L. González Mena, I. Pagonabarraga, and E. Corvera Poiré, “Flow along elastic networks subject to pulsatile forcing” (in preparation).
- ⁴⁰T. Gervais, J. El-Ali, A. Günther, and K. F. Jensen, “Flow-induced deformation of shallow microfluidic channels”, *Lab on a Chip* 6, 500-507 (2006).
- ⁴¹J. Flores, J. Alastruey, and E. Corvera Poiré, “A novel analytical approach to pulsatile blood flow in the arterial network”, *Annals of Biomedical Engineering* 44, 3047-3068 (2016).
- ⁴²M. E. Martinez-Perez, A. D. Hughes, A. V. Stanton, S. A. Thorn, N. Chapman, A. A. Bharath, and K. H. Parker, “Retinal vascular tree morphology: a semi-automatic quantification”, *IEEE Transactions on Biomedical Engineering*, 49, 912-917 (2002).
- ⁴³F. Milde, S. Lauw, P. Koumoutsakos, and M. L. Iruela-Arispe, “The mouse retina in 3D: quantification of vascular growth and remodeling”, *Integrative Biology*, 5, 1426-1438 (2013).

- ⁴⁴K. Ley, A. R. Pries, and P. Gaehtgens, “Topological structure of rat mesenteric microvessel networks”, *Microvascular research*, 32, 315-332 (1986).
- ⁴⁵J. R. Less, T. C. Skalak, E. M. Sevick, and R. K. Jain, “Microvascular architecture in a mammary carcinoma: branching patterns and vessel dimensions”, *Cancer research*, 51, 265-273 (1991).
- ⁴⁶P. Reymond, F. Merenda, F. Perren, D. Rüfenacht, and N. Stergiopoulos, “Validation of a one-dimensional model of the systemic arterial tree”, *American Journal of Physiology - Heart and Circulatory Physiology* 297, 1 (2009).
- ⁴⁷I. F. Stuart, “Fisiología Humana”, 7th ed. Interamericana McGraw-Hill, España (2003). (Spanish translation from “Human Physiology”, McGraw-Hill N.Y. 7th ed., 2002).
- ⁴⁸C. D. Murray, “The Physiological Principle of Minimum Work: I. The Vascular System and the Cost of Blood Volume”, *Proceedings of the National Academy of Sciences of the United States of America* 12, 207-214 (1926).
- ⁴⁹T. F. Sherman, “On connecting large vessels to small. The meaning of Murray’s law”, *The Journal of General Physiology* 78, 431-453 (1981).
- ⁵⁰K.A. McCulloh, J.S. Sperry, and F.R. Adler, “Water transport in plants obeys Murray’s law”, *Nature* 421, 939–942 (2003).
- ⁵¹W. W. Nichols, M. F. O’Rourke, and C. Vlachopoulos, “Theoretical, Experimental and Clinical Principles”, New York: Arnold/Oxford University Press (1998).
- ⁵²N. Xiao, J. Alastruey, and A. C. Figueroa, “A systematic comparison between 1-D and 3-D hemodynamics in compliant arterial models”, *International Journal of Numerical Methods in Biomedical Engineering* 30, 204-231 (2014).

Appendix D

We thank the Associate Editor for the final review of the manuscript. We have implemented all his suggestions except for the one mentioning the pulse wave velocity.

Page 3, use referencing style as per journal []

Done

Page 3, line 50 `may permit to better tailor the' -> `permit better tailoring of the'

Done

Page 3, line 55 use of identical is ambiguous - are the daughter vessels identical to the parent as well as to each other?

Done

Page 4, line 15 `done' -> ``carried out on'

Done

Page 4, line 28 - do not use references in a sentence. Several other instances through the manuscript.

Done. Several instances were altered throughout the document.

Page 5. Avoid use of bullet points.

Done

Page 6. Why are you using the pulse wave velocity? Is this not just a wavespeed?

Equation (3) estimates the vessel Young modulus for an artery. In physiology the pulse wave velocity is the velocity at which the blood pressure pulse propagates through a vessel and it is a property commonly measured in the clinic. Therefore, we believe that in the context of the article, it is the correct expression to include.

Page 7 Line 24 should read `Articles in the literature that classify normal vascular networks consider only bifurcations.

Done

Page 8, line 33, should the q's not have hats as in Eq (9)

Yes. Done.

Page 8, use \langle and \rangle instead of $<$ and $>$

Done.